# Defective transcription elongation in a subset of cancers confers immunotherapy resistance

Vishnu Modur [1], Navneet Singh[1], Vakul Mohanty[2], Eunah Chung[3,4], Belal Muhammad [1], Kwangmin Choi[1], Xiaoting Chen[5], Kashish Chetal[6], Nancy Ratner[1], Nathan Salomonis[6], Matthew T. Weirauch [3,5,6], Susan Waltz[7], Gang Huang[8], Lisa Privette-Vinnedge [9], Joo-Seop Park [3,4], Edith M. Janssen[10] & Kakajan Komurov [1,6,11]

The nature and role of global transcriptional deregulations in cancers are not fully understood. We report that a large proportion of cancers have widespread defects in mRNA transcription elongation (TE). Cancers with TE defects (TE$^{deff}$) display spurious transcription and defective mRNA processing of genes characterized by long genomic length, poised promoters and inducible expression. Signaling pathways regulated by such genes, such as pro-inflammatory response pathways, are consistently suppressed in TE$^{deff}$ tumors. Remarkably, TE$^{deff}$ correlates with the poor response and outcome in immunotherapy, but not chemo- or targeted therapy, -treated renal cell carcinoma and metastatic melanoma patients. Forced pharmacologic or genetic induction of TE$^{deff}$ in tumor cells impairs pro-inflammatory response signaling, and imposes resistance to the innate and adaptive anti-tumor immune responses and checkpoint inhibitor therapy in vivo. Therefore, defective TE is a previously unknown mechanism of tumor immune resistance, and should be assessed in cancer patients undergoing immunotherapy.

[1] Division of Experimental Hematology and Cancer Biology, Cancer and Blood Diseases Institute, Cincinnati Children's Hospital Medical Center (CCHMC), Cincinnati 45229 OH, USA. [2] University of Cincinnati Graduate Program in Systems Biology and Physiology, Cincinnati 45267 OH, USA. [3] Division of Developmental Biology, CCHMC, Cincinnati 45229 OH, USA. [4] Division of Pediatric Urology, New York, NY, USA. [5] Center for Autoimmune Genomics and Etiology, CCHMC, Cincinnati 45229 OH, USA. [6] Division of Biomedical Informatics, CCHMC, Cincinnati 45229 OH, USA. [7] Departments of Cancer Biology and Research Service, University of Cincinnati and Cincinnati Veteran's Hospital Medical Center, Cincinnati 45267 OH, USA. [8] Division of Pathology, CCHMC, Cincinnati 45229 OH, USA. [9] Division of Oncology, CCHMC, Cincinnati 45229 OH, USA. [10] Division of Immunobiology, CCHMC, Cincinnati 45229 OH, USA. [11] Division of Human Genetics, CCHMC, Cincinnati 45229 OH, USA. These authors contributed equally: Vishnu Modur, Navneet Singh, Vakul Mohanty. Correspondence and requests for materials should be addressed to K.K. (email: Kakajan.Komurov@cchmc.org)

Alternative mRNA expression either through differential mRNA splicing, alternative promoter or end-site usage contribute to the complexity of genome regulation. Human cancers, in addition to genomic changes, are also abundant in widespread aberrant alternative transcription events that aid in the tumorigenic process[1]. For example, widespread 3′ shortening of untranslated regions (UTRs) in cancers due to alternative poly-adenylation has been shown to allow tumor cells to escape miRNA-mediated repression of oncogenic pathways[2,3]. In addition, genome-wide alterations in alternative mRNA transcription and intron retention have been observed to frequently activate oncogenes or inactivate tumor suppressor genes[4–7]. Interestingly, although somatic mutations in splicing factors (e.g. U2AF1, SF3B1) and some epigenetic modifiers (SETD2) have been shown to lead to genome-wide alterations in the alternative transcription[1,8], many of the widespread transcriptional defects, such as genome-wide 3′-UTR shortening[3] and intron retention[6,7] mentioned above, do not correlate with any somatic mutations[7,9]. Therefore, transcriptional and epigenetic defects beyond somatic mutations can play key roles in the tumorigenic process, which highlight the need to investigate these non-genetic oncogenic events further. However, although much has been learnt on the molecular mechanisms of transcription and post-transcriptional mRNA processing, the nature, mechanisms, and clinical consequences of their aberrations in cancers are not fully understood.

The mRNA sequencing datasets from The Cancer Genome Atlas (TCGA) provide an unprecedented opportunity to interrogate aberrant transcription events in human cancers and assess their correlation with clinical parameters. Through extensive pan-cancer analyses of aberrant transcriptional events, we find that a significant portion of all human cancers are characterized by highly abnormal genome-wide transcriptional profiles suggestive of defective RNA Polymerase II (RNAP II) elongation function. Importantly, this phenotype, which we termed defective transcription elongation (TE$^{deff}$), has a specific profound effect on the expression of stimulus-responsive long genes and associated pathways. As such, many of the key inflammatory pathway genes, such as TNF/NF-κB and interferon/STAT signaling, which are heavily regulated at the level of transcription elongation[10,11], are specifically suppressed in tumors with TE$^{deff}$. Thus, TE$^{deff}$ leads to impaired response to pro-inflammatory death stimuli, resistance to immune-mediated attacks and, consequently, to immunotherapy resistance in the clinic. We suggest that TE$^{deff}$ is a previously unknown epigenetic mechanism of silencing of key inflammatory response pathways, and of resistance to immunotherapy.

## Results

**Some cancers have high expression of truncated isoforms.** To dissect regulatory variations of isoform-level alternative expression from gene-level variations, we defined several metrics to score the extent of each gene's regulation at the level of transcript isoform switching (see Supplementary Fig 1A and legend). Pan-cancer analyses (Supplementary Table 1) of isoform- vs. gene-level expression variations of genes using these metrics revealed that a subset of genes is characterized by high isoform-level variance in expression, indicating explicit regulation at the level of alternative transcription (AT genes) (Supplementary Fig 1A). The AT genes were enriched for those involved in mRNA processing, innate inflammatory signaling, and chromatin remodeling (Supplementary Fig 1B), and were highly similar across the cancers of different tissue types (Supplementary Fig 1C), indicating that transcript isoform-level regulation is an inherent property of this class of genes. Interestingly, an all-against-all expression correlation analysis of the transcript isoforms of AT genes revealed

that most of them displayed a bimodal expression pattern (Supplementary Fig 1D), such that, while the majority of tumors predominantly expressed the full-length canonical isoforms, a subset of tumor, but not adjacent normal tissue, samples preferentially expressed the shorter isoforms predicted to code for truncated proteins for these genes (Fig. 1a, b and Supplementary Fig 1E). We verified that this pattern is not an artifact of the RNAseq processing methods, sample quality or batch effect (Supplementary Fig 2A, B). Applying the same criteria (see "Scoring TE$^{deff}$" in Methods) to other cancers revealed that this pervasive transcript shortening is a widespread phenotype in all cancers we have analyzed, observed in more than 20% of almost every cancer type (Fig. 1c).

**Defective and spurious transcription in a subset of cancers.** To gain deeper insight into the transcriptional aberrations in the tumors with the widespread transcript shortening (TS), we performed an analysis of differential exon expression in TS+ (i.e. those that have TS) vs. TS- samples using the RNAseq (polyA-selected) datasets in TCGA. The genome-wide differential exon expression heatmaps showed that a large proportion of all measured genes had a widespread significant loss in the expressions of their gene body exons and a significant increase in the expression of the 3′-terminal exons (Fig. 1d), with still many genes overall overexpressed, a pattern that was reproduced in the TS+ tumors of many cancers (Supplementary Fig 3A).

The exon-level expression pattern in Fig. 1d suggests defects in the transcription of gene body exons, and preferential spurious transcription of the terminal exons for a large number of genes (class I genes), although still many genes were overexpressed in these tumors (class II genes) (see Fig. 1d) (see Supplementary Table 2 for Class I and II genes). To rule out technical artifacts from polyA-selected RNA sequencing that could elicit this pattern, we carried out a similar analysis using Affymetrix Exon array data in glioblastoma (GBM), lung squamous carcinoma (LUSC) and ovarian cancer (OV) samples (exon array data are only available in these three). Importantly, the mRNAs measured in exon arrays are not polyA-selected, and thus offer a whole-transcriptome view of the mature as well as nascent transcripts, rather than focusing on mature polyA-ed mRNAs. Strikingly, in accordance with the observed patterns with RNAseq, we observe a consistent and significant decrease in the usage of exons within the gene bodies (Fig. 1e and Supplementary Fig 3B). However, the exon array profile also displayed a sharp peak around the transcription start site (TSS) in TS + tumors, especially in the class I genes (Fig. 1e and Supplementary Fig 3C), which gradually disappeared in ~1 kB after TSS (Fig. 1f). Since this peak is not observed in the polyA-selected RNAseq patterns from the same samples (see Fig. 1d), these short transcripts are likely not poly-adenylated. Interestingly, this pattern resembles the TSS-associated short capped RNAs (tssRNAs) produced by stalled RNAP II during elongation arrest, which are also not poly-adenylated[12,13], suggesting widespread defects in the elongation of nascent transcripts by RNAP II into the gene body in the TS+ tumors.

Again consistent with the polyA RNAseq pattern, there is a sharp peak in the usage of the most terminal exons in TS+ tumors (Fig. 1e), supporting extensive spurious transcription initiation. This is consistent with the prior findings that the perturbation of transcription elongation leads to spurious intragenic transcription from 3′ sites[14,15]. Based on this and later observations presented below, we have named the TS phenotype presented above as defective transcription elongation (TE$^{deff}$). For the rest of the manuscript, we will refer to tumors with TE$^{deff}$ as TE$^{deff}$ tumors, and the rest as TE$^{prof}$, for TE-proficient, although

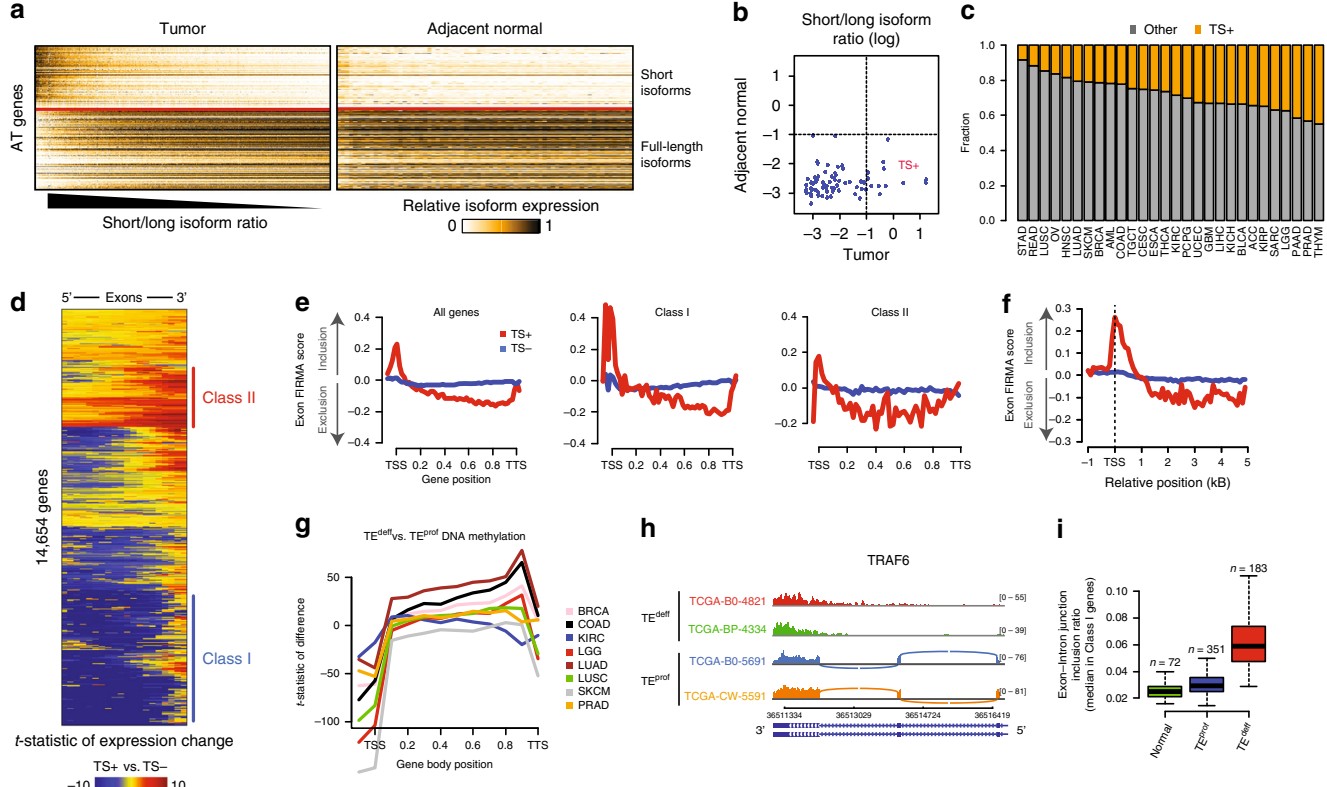

**Fig. 1** A subset of cancers display transcription elongation defects (TE$^{deff}$). **a** Heatmap of relative expression of short and long transcript isoforms of AT genes (see Supplementary Fig 1a–d) in the tumor and adjacent normal samples of clear cell renal carcinomas (KIRC). Samples (columns) are ordered by average short/long isoform ratio as shown. **b** Short-to-long isoform expression ratios (log) in the matching tumor and normal samples in KIRC. Cutoff for TS is defined as average log ratio >−1. **c** Distribution of TS+ and TS− samples in different cancers. **d** Heatmap of exon-level expression difference in TS+ vs. TS− tumors from KIRC. Rows show genes and columns represent exon bins ordered from 5′ to 3′ (see Methods). Class I and II genes are highlighted. See Supplementary Fig 3a for other cancers. **e** Differential exon inclusion/exclusion patterns in all (left), Class I (middle) and Class II (right) genes in TS+ and TS− samples based on the Affymetrix Exon Array data in lung squamous cell carcinoma (LUSC). Y-axes show median FIRMA scores for exons (reflecting inclusion [>0] or exclusion [<0]) in the given gene position on the x-axis. See Supplementary Fig 3b for ovarian carcinoma (OV) and glioblastoma (GBM). **f** Same as in **e**, with the x-axis showing the distance in kBs from the TSS. **g** Differential DNA methylation patterns in TE$^{deff}$ samples (TS+) relative to TE$^{prof}$ (TS−) samples in the indicated cancers based on 450k methylome array data. The y-axis shows the t-statistic of difference of the respective β-values. TSS transcription start site, TTS transcription termination site. **h** Coverage plot of RNA sequencing reads from two TE$^{deff}$ and two TE$^{prof}$ samples from a 3′-most region of *TRAF6* gene. Note "bleeding" of reads into the intronic regions and lack of exon-exon junction reads in TE$^{deff}$ samples. Sashimi plots of the full gene are shown in Supplementary Fig 4A. **i** Boxplot of exon–intron and intron–exon junctions (ratio to exon–exon junctions) in Class I genes in Normal, TE$^{prof}$, and TE$^{deff}$ KIRC samples. Boxplots: middle line: median, boxed areas extend from the first to third quartile; whiskers show 1.5 x inter-quartile range from the first (bottom) or third (top) quartile

we recognize that the TE$^{prof}$ tumors may still have other transcriptional defects (e.g. shortened 3′-UTRs, etc).

**Alterations in DNA methylation in TE$^{deff}$.** Epigenetic modifications, such as histone and DNA methylations, along the gene bodies are often closely correlated with the transcription of the corresponding sequences[16,17]. Therefore, we tested if TE$^{deff}$ tumors are associated with the DNA methylation patterns reflective of their aberrant transcription. Interestingly, we found that TE$^{deff}$ tumors had a significant decrease in the DNA methylation around the transcription start sites (TSS), an increase within the gene bodies and another dip around the transcription termination sites (TTS) (Fig. 1g). Given that DNA methylation is often restrictive of transcription[16], this pattern in DNA methylations is in a high accordance with the aberrant transcriptional patterns observed in TE$^{deff}$ tumors (see Fig. 1d–f), suggesting coordinated epigenetic and transcriptional defects in TE$^{deff}$ tumors.

**mRNA splicing and spurious transcription defects in TE$^{deff}$.** The elongating RNAP II serves as the assembly platform for various mRNA splicing and processing machineries, and its perturbation has widespread effects on many aspects of mRNA homeostasis, including splicing and post-transcriptional processing[18,19]. Accordingly, we also noticed that many genes in TE$^{deff}$ tumors, in addition to the loss of gene body exon expression, seemed to have aberrant exon definition and splicing as well as spurious and antisense transcription (Fig. 1h and Supplementary Fig 4). To quantify aberrant exon definition and splicing events, we measured genome-wide intron–exon and exon–intron junction retention in mRNAs (see Methods), and found that TE$^{deff}$ tumors had indeed genome-wide splicing defects, specifically in class I genes, at a significantly higher rate compared to normal and TE$^{prof}$ samples (Fig. 1i).

**TE$^{deff}$ does not correlate with frequent somatic mutations.** Interestingly, TE$^{deff}$ did not consistently correlate with any of the frequent (i.e. >5% rate) somatic mutations in cancers, indicating

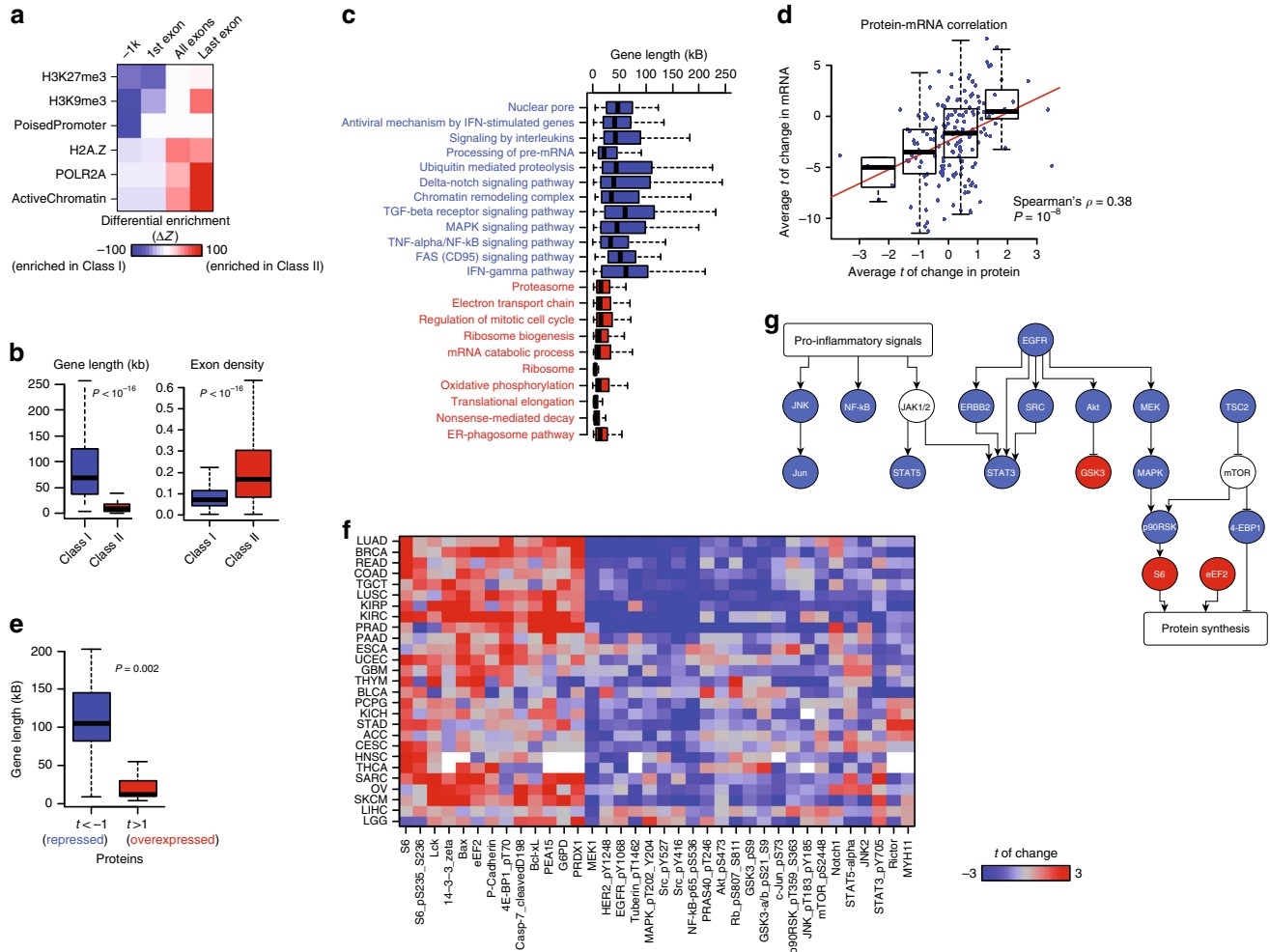

**Fig. 2** TE$^{deff}$ impairs the expression of stimulus-responsive genes and pathways. **a** Differential enrichment of Class I and II genes for the gene features defined in the Roadmap Epigenomics datasets. **b** Genomic lengths (left) and exon density (ratio of full mRNA/genomic length) profiles (right) of Class I and II genes (*P*: t-test). **c** Genomic length distributions of constituent genes in the Class I- and Class II-enriched (*P* < 0.0001, hypergeometric test) pathways (colored blue and red, respectively). **d** Scatter plot of average t-statistic of change in mRNA (*y*-axis) and corresponding total protein (*x*-axis) in TE$^{deff}$ vs. TE$^{prof}$ samples across cancers. **e** Gene lengths of proteins that are repressed (average t-statistic <−1) or overexpressed (t-statistic >1) in TE$^{deff}$ cancers. **e** A heatmap of some of the most consistent protein-level changes in TE$^{deff}$ vs. TE$^{prof}$ samples in the indicated cancer types (*P*: t-test). **f** A diagram of representative pathways from commonly repressed and overexpressed phospho- and total proteins in F (blue: repressed, red: overexpressed). Compare the pathways with the mRNA-level profile in **c**. Boxplots:: middle line: median, boxed areas extend from the first to third quartile; whiskers show 1.5x inter-quartile range from the first (bottom) or third (top) quartile

that it might be largely an epigenetic phenomenon, akin to previously reported transcriptional aberrations (e.g. 3′-UTR shortening)[1,3,4,7].

**Class I genes are enriched for stimulus-responsive genes.** Given that the effect of TE$^{deff}$ is not uniform across the genome, we asked if class I (defective transcription and splicing) and class II (overexpressed) genes are structurally and functionally distinct. For this purpose, we first analyzed the enrichment of these classes of genes for specific genomic regulatory elements using the published consortia datasets[20]. Interestingly, while class II genes primarily represented actively transcribed genes, class I genes were characterized by "poised" promoters (Fig. 2a). Genes with poised promoters are characterized by stalled RNAP II, condition-specific (i.e. inducible) expression and are primarily regulated at the level of transcription elongation (i.e. pause release)[11,21]. Consistent with inducible genes[22], class I genes were also significantly longer than the class II genes, and contained longer introns (i.e. lower full mRNA/genomic length ratio)

(Fig. 2b). These observations support the notion that TE$^{deff}$ is a phenotype of defective transcription elongation that primarily affects genes with inducible promoter architectures that are regulated at the level of pause-release.

**TE$^{deff}$ suppresses pathways regulated by class I genes.** Genes with rapidly inducible poised promoters are enriched for stimulus-responsive pathways, such as NF-κB, MAP kinase, JAK/STAT, JNK, and TGFβ[10,23]. As such, class I genes were enriched for various regulatory pathways, such TNF-α/NF-κB, Delta-Notch, TGFβ, MAP kinase, EGFR, and interferon; whereas class II genes were primarily enriched for homeostatic processes: translation initiation and elongation, proteasome and mitochondrial oxidation (Fig. 2c). Interestingly, the genomic length distribution of genes in these respective pathways were in accordance with the enrichment of long genes among class I genes (Fig. 2c). Importantly, analyses of the reverse-phase protein array (RPPA) datasets from TCGA revealed that protein-level changes in tumors with and without TE$^{deff}$ were highly consistent

with their mRNA-level changes (Fig. 2d). In addition, while proteins that were repressed in TE$^{deff}$ tumors were encoded by long genes, those that were overexpressed were encoded by short genes (Fig. 2e), indicating that mRNA-level defects in TE$^{deff}$ are also observed at the protein level. Moreover, the most consistent phospho- and total protein changes in TE$^{deff}$ tumors in RPPA data (Fig. 2f) revealed pathway activation profiles that were in close agreement with the pathway enrichment profiles of class I and II genes (see Fig. 2c); where protein synthesis pathways proteins, such as eEF2, ribosomal S6, and EIF4BP1, were overexpressed, while pro-inflammatory pathway proteins, such as NF-κB, STAT3, STAT5, and JNK, as well as EGFR/MAP kinase signaling, are consistently repressed, in TE$^{deff}$ tumors (Fig. 2g).

**TE$^{deff}$ is detectable in an independent tumor cohort.** To test if the TE$^{deff}$ phenotype can be identified in an independent tumor cohort, we performed RNA sequencing of a panel of 12 advanced clear-cell renal cell carcinoma (ccRCC) tissues (TE$^{deff}$ was highly abundant in advanced ccRCC cases in TCGA, see Fig. 1c) from the University of Cincinnati Tumor Bank. Applying the same pipeline as used for TCGA datasets, we identified 3 samples (25%) to have high short/full-length isoform expression for AT genes and highly TE$^{deff}$-like transcriptomic signature (Supplementary Fig 5). In addition, these samples displayed loss of gene body exon expression, especially in the long genes, and had pathway enrichment profiles similar to the TE$^{deff}$ samples in TCGA (Supplementary Fig 5). Therefore, TE$^{deff}$ is a true phenomenon in cancers, and is associated with genome-wide defects in transcription elongation and mRNA processing.

**TE$^{deff}$ is present in some cancer cell lines.** To identify in vitro models of TE$^{deff}$ for mechanistic studies, we interrogated the Cancer Cell Line Encyclopedia (CCLE) RNAseq datasets[24]. We found two breast cancer (UACC812, MDA-MB415) and two leukemia cell lines (RS411, HL60) to have global transcript shortening and intron retention profiles highly consistent with TE$^{deff}$ in cancer tissues (Fig. 3a). In addition, the differential exon- (Supplementary Fig 6) and gene-level (Fig. 3a) transcriptomic signatures of these lines strongly resembled the respective TE$^{deff}$ signatures from TCGA samples, which we verified by independent in-house RNA sequencing (Fig. 3b, c), where the correlation of TE$^{deff}$-specific transcriptomic signature in cell lines with those in TCGA samples reached $\rho = 0.38$ (Spearman's, $P < 10^{-16}$). The spuriously transcribed genes in this dataset were significantly enriched for the Class I genes from the TCGA datasets, and were, accordingly, significantly longer (Fig. 3c, d), indicating that these lines are bona fide TE$^{deff}$ cells.

To measure genome-wide RNAP II occupancy and spurious transcription events in TE$^{deff}$ lines, we performed genome-wide chromatin immunoprecipitation and sequencing (ChIP-seq) using antibodies against total and Ser5-phosphorylated RNAP II. Phosphorylation of RNAP II at Ser5 position at its C-terminal domain (CTD) is a marker of transcription initiation sites. UACC-812 cells (TE$^{deff}$) had increased total RNAP II occupancy at the transcription start sites (TSS), and an increased occupancy at the transcription termination sites (TTS), compared to T47D cells, which are TE$^{prof}$ (Fig. 3e, f). On the other hand, positioning of total RNAP II along the Class I and II genes reflected their expression in TE$^{deff}$ cells, with Class I genes having increased TSS and reduced gene body occupancy, consistent with RNAP II elongation defect on these genes, while Class II genes showed increases in both (Supplementary Fig 7). Interestingly, the distribution of Ser5-phosphorylated RNAP II showed increased occupancy within the gene bodies, especially closer to TTS, in UACC-812 cells (Fig. 3e, f), suggesting increased spurious

transcription initiation from intragenic cryptic promoters, which was also prominently observed in Class II genes (Supplementary Fig 7). These observations strongly suggest that TE$^{deff}$, in addition to TE defects in Class I genes, are also associated with widespread spurious intragenic transcription.

In accordance with defective RNAP II function, these lines showed diminished levels of total RNAP II levels, and phosphorylation at CTD Ser2 (marker of transcription elongation) and Ser5 positions. In addition, Cyclin T1 (CCNT1), the cyclin partner of the p-TEFb complex involved in RNAP II C-terminal phosphorylation at Ser2 position, required for RNAP II pause-release, is also significantly suppressed, along with histone H3 acetylation and trimethylation at lysine 36 (H3K36me3), also a marker of transcription elongation (Fig. 3e). In addition, these lines showed severe mRNA processing defects, with highly increased ratios of improperly capped and poly-adenylated mRNAs (Fig. 3f), which is expected in cells with defective RNAP II elongation[19] (see below). These results show that TE$^{deff}$ cells have severe defects in the fidelity of RNAP II-mediated mRNA transcription and processing.

**TE$^{deff}$ correlates with resistance to immunotherapy in clinic.** Next, we asked if TE$^{deff}$ correlated with clinical outcome. TE$^{deff}$ predicted poor survival in some cancers, most significantly in clear cell renal cell carcinomas (KIRC), where TE$^{deff}$ was enriched among advanced cases (Supplementary Fig 8). However, stratifying patients based on their therapy modalities in KIRC revealed a striking difference in the survival outcome of TE$^{deff}$ patients. While TE$^{deff}$ patients treated with immunotherapy (primarily interleukin and interferon) had significantly poor survival, those treated with targeted therapy had a significantly better outcome (Fig. 4a, b), suggesting specific resistance to immunotherapeutic drugs. To further test the correlation of TE$^{deff}$ with immunotherapy response, we analyzed metastatic melanoma (SKCM) cases in TCGA that received immunotherapy. Strikingly, TE$^{deff}$ melanoma patients treated with cytokine (interleukin, interferon) or immune checkpoint inhibitor (ipilimumab, pembrolizumab, nivolumab) therapy had significantly poor outcome compared to patients without TE$^{deff}$ (Fig. 4c). To further confirm this observation in a better-controlled immunotherapy cohort, we analyzed ipilimumab (anti-CTLA4 immune checkpoint inhibitor)-treated melanoma patient cohort from Van Allen et al.[25] ($n = 42$ patients). Consistent with our findings in TCGA cohorts, patients with LTF-like transcriptional defects (see Methods) had significantly higher rate of therapy resistance and poor progression-free and overall survival (Fig. 4d).

We also tested this phenomenon in the recent cohort from Hugo et al.[26], who obtained RNAseq data from 27 melanoma patients treated with anti-PD1 therapy (pembrolizumab). Here too, TE$^{deff}$ correlated with treatment resistance, and shorter overall survival compared to other patients (Fig. 4e). These observations suggest that TE$^{deff}$ confers a generic resistance to both cytokine and checkpoint inhibitor immunotherapies.

In the original studies of Van Allen et al. and Hugo et al., they identified increased nonsynonymous mutational burden (NMB) and tumor T-cell infiltration (TIL) as strong correlates of therapy response and prognostic outcome. Importantly, TE$^{deff}$ tumors did not have lower mutational load or tumor infiltration by lymphocytes (TIL) in these datasets (Supplementary Fig 9A–D), suggesting that the effect of TE$^{deff}$ on immunotherapy response is not due to its potential effect on TIL or mutational burden. This further suggests that TE$^{deff}$ and TIL, or mutational burden, could be combined for a greater separation of patients who may or not benefit from immunotherapy. Indeed, TE$^{deff}$ strongly synergized with these biomarkers in predicting clinical benefits in the

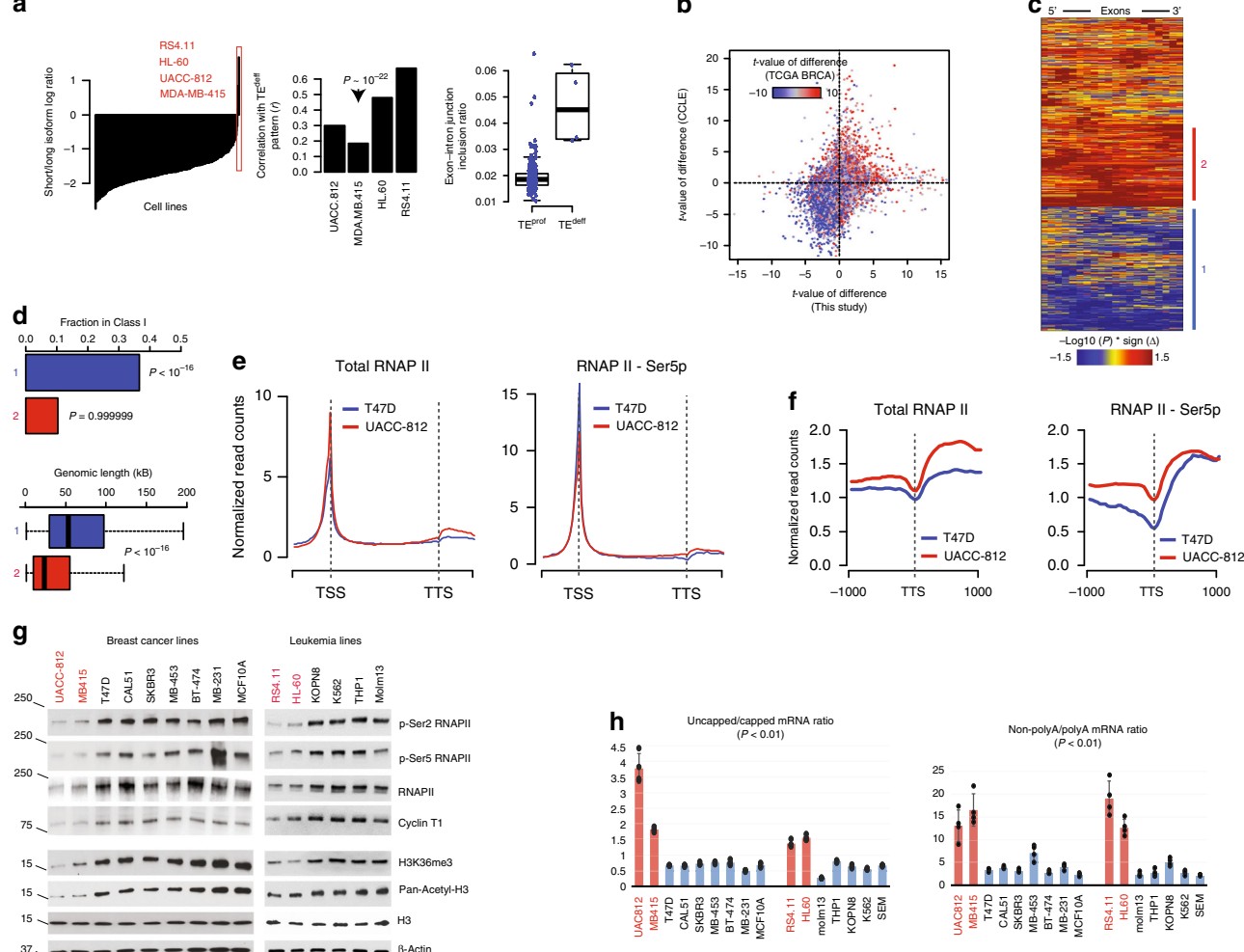

**Fig. 3** TE$^{deff}$ is observed in some cancer cell lines and correlates with defects in mRNA transcription, processing, and export. **a** Left: cell lines are ordered based on the average short/long isoform ratio for the AT genes. Four with the highest scores are highlighted. Middle: correlation of the transcriptional patterns (every gene is z-normalized to the rest of the cell lines) of candidate TE$^{deff}$ cell lines with the average TE$^{deff}$ signature (average t-statistic of TE$^{deff}$ vs. TE$^{prof}$ expression difference for every gene). The *p*-value of correlation (Spearman) for the cell line with the lowest correlation is shown. Right: quantification of intron–exon junction inclusion ratio in Class I genes (same as in Fig. 1h) in TE$^{prof}$ and TE$^{deff}$ cell lines. **b** Three-way correlation of TE$^{deff}$-specific transcriptomic signatures (t-values of difference in TE$^{deff}$ vs. TE$^{prof}$ cells for every gene) obtained for breast cancer cell lines based on RNAseq data from CCLE (y-axis), based on our independent RNAseq (x-axis) and based on RNAseq data from TCGA breast cancer (BRCA) dataset (color-mapped based on the color key). **c** Differential exon expression heatmap (see Fig. 1c) of TE$^{deff}$ and TE$^{prof}$ breast cancer lines obtained from our independent RNAseq data. Genes with spurious and overexpressed expression patterns are highlighted (blue and red, respectively). **d** Enrichment of the spuriously transcribed (blue) and overexpressed (red) genes in **c** for the Class I genes from the tissue samples. The *p*-values reflect one-sided hypergeometric distribution, i.e. closer to 0 indicates enrichment, closer to 1 indicates depletion. **e** Genome-wide distribution plots of total and phospho-Ser5 RNAP II in a TE$^{deff}$ (UACC-812) and a TE$^{prof}$ (T47D) cell line. **f** Genome-wide distribution of total and phospho-Ser5 RNAP II around the transcription termination site (TTS) regions of genes in TE$^{deff}$ and TE$^{prof}$ cells. **g** Immunoblotting of indicated RNAP II and histone marks in the indicated cell lines. TE$^{deff}$ cells are highlighted in red. **h** Ratios of 5'-uncapped to 5'-capped (left) and 3'-non-polyadenylated to 3'-polyadenylated (right) mRNA concentrations after rRNA depletion in the indicated cell lines. *P*: difference between TEdeff and TEprof cell line values (*t*-test). Error bars: standard deviation based on four technical replicates

respective cohorts, with TE$^{deff}$/TIL-low and TE$^{deff}$/NMB-low patients having the worst prognosis, and TE$^{prof}$/TIL-high or TE$^{prof}$/NMB-high patients having the best (Fig. 4f, g).

**TE$^{deff}$ impairs IFN and TNF pathway signaling.** Tumor cell signaling competency through innate inflammatory response pathways, such as interferon (IFN)/JAK/STAT, NF-κB, Fas and antigen presentation, has been shown to be essential for immunotherapy efficacy in the experimental and clinical settings[27–32]. In addition, recent genome-wide screening studies provided strong evidence for the importance of these pathways in the

response to anti-tumor immune attack[33,34]. The pro-inflammatory response genes are highly stimulus-responsive, and are strongly regulated at the level of transcription elongation[10,11,23]. Accordingly, the expression of many key genes in the IFN, NF-κB, and Fas/JNK/Caspase 8 pathways identified in these studies is impaired in TE$^{deff}$ cancers and cell lines (Fig. 5a, see Fig. 2b–f, and Supplementary Fig 9E for the Hugo et al and Van Allen et al datasets). In addition, their mRNAs were not properly post-transcriptionally processed (i.e. poly-adenylated and 5'-capped) in the TE$^{deff}$ cell lines, and were largely retained in the nucleus (Fig. 5b). Consequently, TE$^{deff}$ cell lines had a

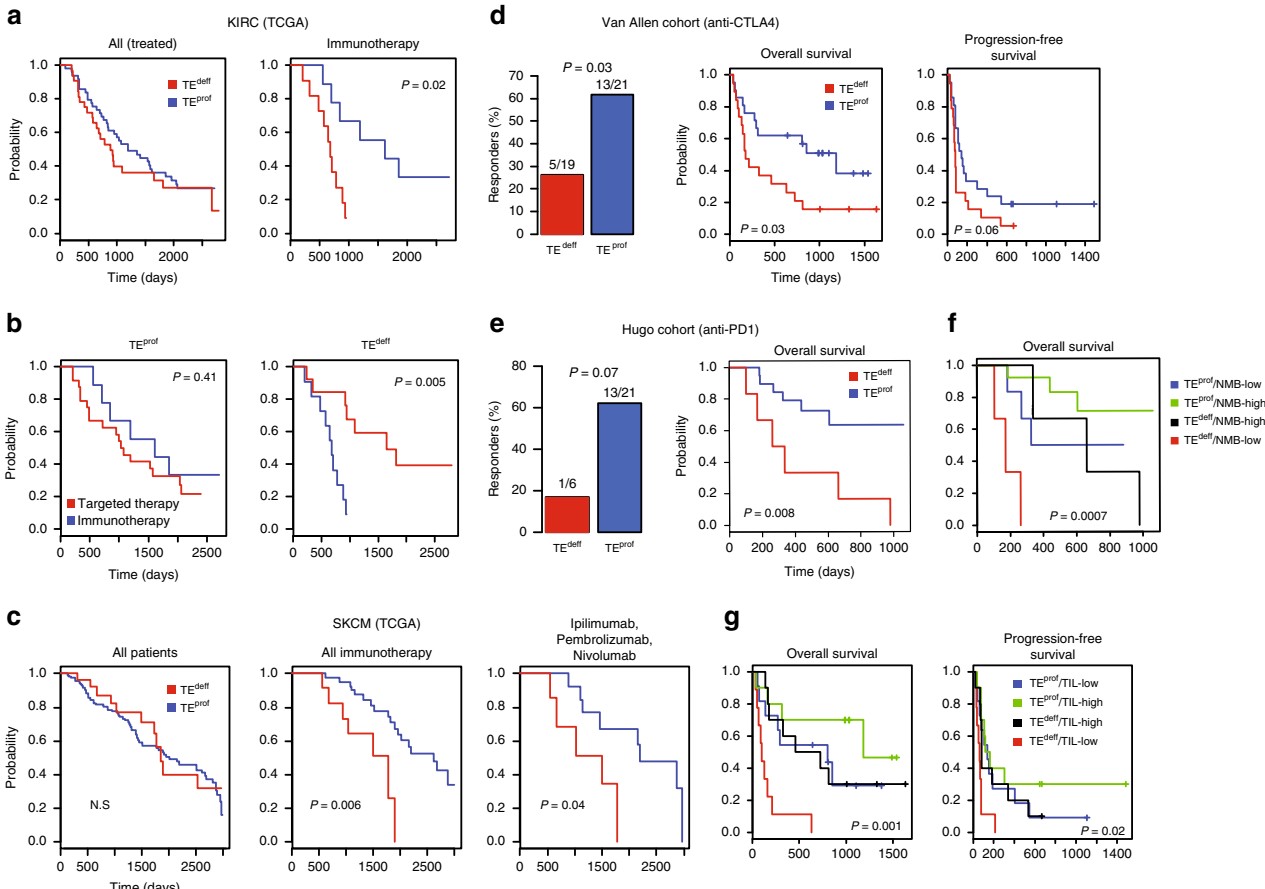

**Fig. 4** TE$^{deff}$ correlates with clinical resistance to immunotherapy. **a** Kaplan–Meier survival curves of TE$^{deff}$ and TE$^{prof}$ KIRC patients. Left: all patients with treatment data, right: those that received immunotherapy (IL2 and IFN), **b** Kaplan–Meier survival curves of TE$^{prof}$ and TE$^{deff}$ patients treated with immunotherapy or targeted therapy. **c** Same as in **a** in SKCM patients. Middle: patients treated with immunotherapy other than immune checkpoint inhibitors, right: patients that received immune checkpoint inhibitor therapy. **d** Left: percentage of therapy responders among TE$^{deff}$ and TE$^{prof}$ subset of the Van Allen et al cohort, right: progression-free and overall survival curves of TE$^{deff}$ and TE$^{prof}$ samples in this cohort, **e** Left: percentage of responders among TE$^{deff}$ and TE$^{prof}$ subset in the Hugo et al. cohort, right: Kaplan–Meier curves for OS in TE$^{deff}$ and TE$^{prof}$ patients, **f** overall survival curves of patients in the Hugo cohort stratified based on TE$^{deff}$ and the non-synonymous mutational burden (NMB), **g** Kaplan–Meier curves of OS and PFS in the Van Allen cohort stratified according to TE$^{deff}$ and TIL status. The p-values in survival analyses reflect Wald test, and in the barplots Fisher's exact test

significantly reduced expression of these genes at the protein level (Fig. 5c), and had diminished response to the type I and II interferon, as well as TNF-α, stimulation in vitro (Fig. 5d). Importantly, the expression of some key antigen presentation pathway proteins was also significantly reduced, and was not responsive to IFN stimulation (Fig. 5e), predicting poor antigen presentation in TE$^{deff}$ cells, which is a hallmark of evasion of anti-tumor immune attack in vivo[33,34] and of resistance to immunotherapy in the clinic[35]. Moreover, they were more resistant to cell death by the death receptor ligand FasL (Fig. 5f), which is a major mechanism of tumor cell killing by cytolytic lymphocytes.

**Chronic inhibition of RNAP II recapitulates TE$^{deff}$.** To test the causal role of defective TE in the resistance to anti-tumor immune response, we chronically stimulated B16/F10 mouse melanoma cells with sublethal doses (25 nM) of flavopiridol, an inhibitor of the RNAP II elongation factor p-TEFb (Cyclin T/ CDK9). Treatment of cells with flavopiridol at this dose did not significantly perturb the cell cycle profile or population growth (Supplementary Fig 10), suggesting that the off-target effect of flavopiridol on CDKs 1 and 2 is negligible at this concentration in these cells (flavopiridol has ~7–8 fold selectivity towards CDK9 [IC50 ~3 nM] over other CDKs[36]). One week prolonged

treatment of B16/F10 cells with flavopiridol mimicked TE$^{deff}$ in terms of widespread RNAP II phosphorylation, histone modification and mRNA processing defects (Fig. 6a and Supplementary Fig 11A), and led to a transcriptional signature that was highly similar to that of TE$^{deff}$ cancer tissues (gene-by-gene correlation of significant expression changes: Pearson's $r = 0.22$, $P = 10^{-31}$). There was a significant overlap of the chronic flavopiridol-repressed genes with Class I genes, and the overexpressed genes with Class II genes (Fig. 6b, c).

ChIP-seq experiments using antibodies against total RNAP II in these cells showed reduced overall traveling RNAP II ratios (ratio of RNAP II within the gene body to that around the TSS) in chronic flavopiridol-treated cells, where Class I genes were among the most affected (Fig. 6d–f). Importantly, while gene body RNAP II occupancy strongly correlated with mRNA expression (polyA+) output from the respective genes in the parental B16F10 cells, it did significantly less so in chronic flavopiridol-treated cells (Fig. 6g), suggesting spurious, or non-productive, RNAP II activity in these cells. In support of spurious intragenic transcription in these cells, we found increased expression of the middle and 3′-most exons for Jak1 gene in chronic flavopiridol-treated cells in the baseline and upon IFN-γ stimulation (Fig. 6h).

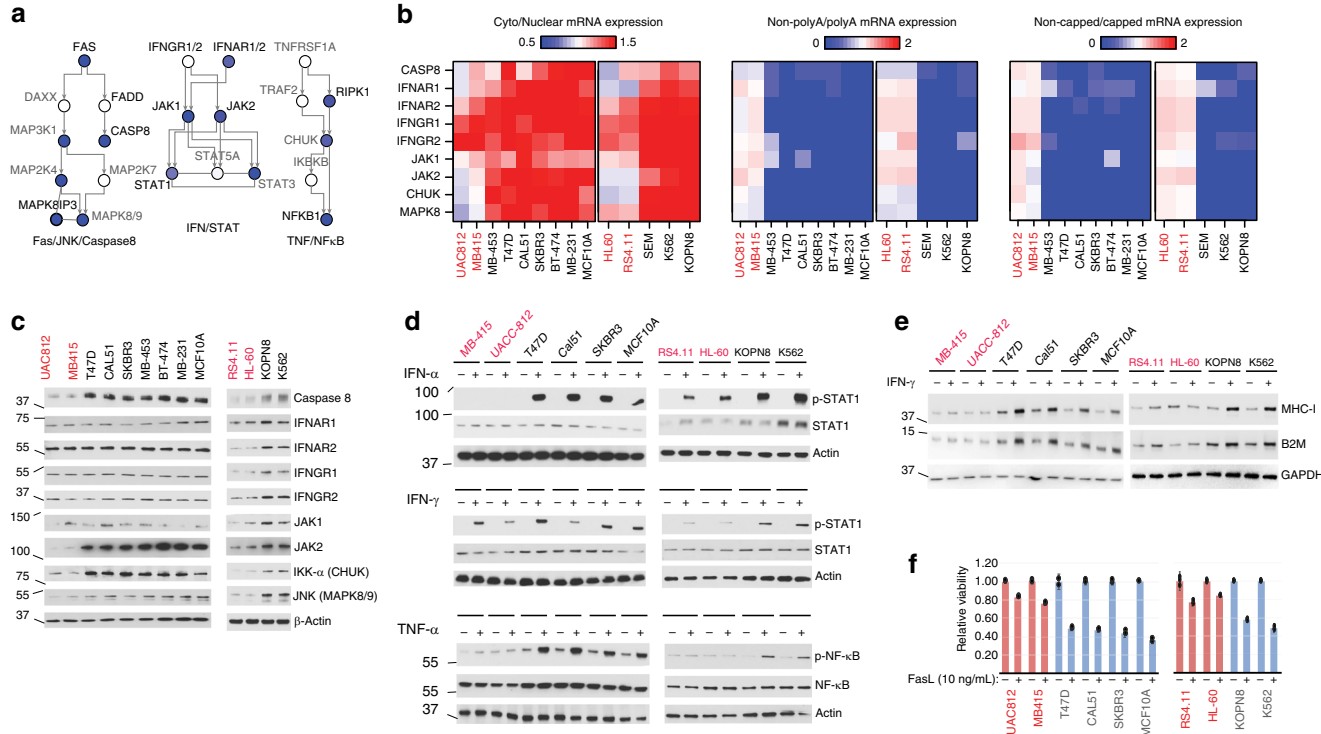

**Fig. 5** TE$^{deff}$ impairs the expression of and signaling through the inflammatory response pathways and antigen presentation. **a** A diagram of pro-inflammatory response pathways relevant to anti-tumor immune surveillance. Blue nodes indicate those that are found among Class I genes. Genes that have been identified as modulators of tumor response to anti-tumor immune attack in Patel et al.[35] and Manguso et al.[34] are indicated with bold labels. **b** Ratios of cytoplasmic to nuclear (left), non-polyadenylated to polyadenylated (middle) and uncapped to capped mRNA levels of the key inflammatory response pathway genes in the indicated cell lines (qPCR). **c** Protein expression of the genes in **b** in the indicated cell lines. **d** TE$^{deff}$ and TE$^{prof}$ cells were stimulated with IFN-α, IFN-γ, and TNF-α for 30 min and the readout measured by immunoblotting. **e** Immunoblots of the expression of MHC-I and B2M proteins in the indicated cell lines at the baseline and upon stimulation with IFN-γ. **f** Indicated cell lines were stimulated with FasL for 24 h and the viability was measured. Error bars: standard deviation based on 3 technical replicates

**Chronic inhibition of RNAP II impairs inflammatory signaling.** Chronic flavopiridol led to specific repression of the key inflammatory response pathway genes at both mRNA and protein levels (Fig. 7a), and imposed resistance to interferon, TNF-α and FasL treatments (Fig. 7b). Similar results were observed in cells with CRISPR-based *CCNT1* knock-out, the cyclin component of the p-TEFb complex (Supplementary Fig 11B), indicating that the impairment of the inflammatory response with flavopiridol treatments are due to on-target effects on the CCNT1/CDK9 activity. Importantly, the stable overexpression of Jak1, a key player in tumor cell response to anti-tumor immune attacks[29,30,33,34], reversed flavopiridol-induced resistance to IFN and FasL treatments (Supplementary Fig 11C, D). However, Jak1 overexpression only reversed the flavopiridol-mediated repression of some, but not all, key genes in the inflammatory pathways (Supplementary Fig 11E), suggesting that only part of the effect of chronic TE$^{deff}$ is due to the suppression of the IFN/Jak pathway. These observations are intriguing as they suggest that the chronic perturbation of transcription elongation is sufficient to elicit the wide range of mRNA homeostatic defects in TE$^{deff}$ cancers, and demonstrate the causal role for the suppression of the inflammatory response pathway genes in the ensuing resistance to anti-tumor immune stimuli.

**TE$^{deff}$ induction confers resistance to T-cell attack.** Consistent with TE$^{deff}$ cell lines, flavopiridol pre-treatment also led to reduced cell surface expression of HLA-I antigen presentation complex, which was also partially rescued by Jak1 overexpression

(Supplementary Fig 12A). To test if TE$^{deff}$ induction by flavopiridol will confer resistance to cytotoxic T-cell (CTL) attack, we co-incubated B16/F10 cells stably overexpressing the ovalbumin (OVA) gene (B16/F10-OVA) with the activated CD8 + CTLs from the spleens of OT-I mice. OT-I mice are transgenic for T-cell receptors that are specific for the OVA$_{257-264}$ epitope[37], and their CD8 + CTLs therefore have selective toxicity to OVA-expressing cells. While the parental B16/F10 cells were not susceptible to OT-I CTL-mediated tumor lysis (due to no OVA expression), B16/F10-OVA cells underwent massive cell death in this system (Fig. 7c). However, B16/F10-OVA cells chronically pre-treated with flavopiridol were highly resistant to CTL-mediated attack (Fig. 7c).

**TE$^{deff}$ induction confers resistance to NKs in vivo.** We asked if chronic flavopiridol-induced TE$^{deff}$ can confer escape from anti-tumor immune attack in vivo. Anti-tumor immune surveillance involves both innate and adaptive immune components;[38–40] therefore, first, we tested if prolonged flavopiridol treatment of B16/F10-OVA cells can confer escape from innate immune-mediated tumor rejection. Natural Killer (NK) cells provide rapid and potent immunity against metastases[38,41], and seeding of B16/F10-OVA cells in the lungs of immune-competent C57BL6 mice following tail vein injection is strongly regulated by NK-mediated tumor rejection[42]. Intriguingly, inhibition of RNAP II elongation by chronic flavopiridol pre-treatment significantly increased the ability of B16/F10-OVA cells for lung seeding compared to control cells (Fig. 7d). Depletion of NK cells before tumor cell

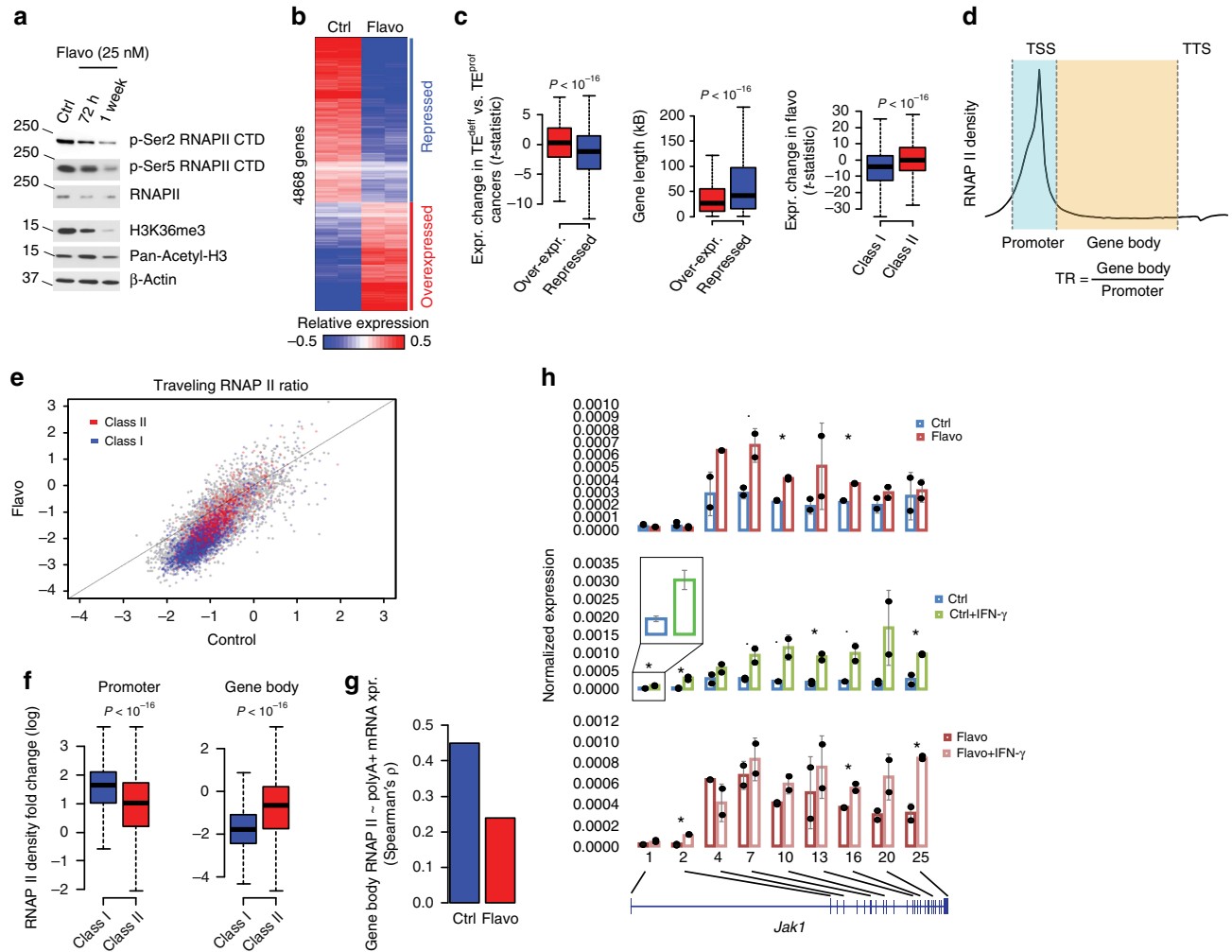

**Fig. 6** Chronic inhibition of TE recapitulates TE$^{deff}$. **a** Immunoblots of indicated histone and RNAP II marks in B16/F10 cells treated with flavopiridol for 72 h or 1 week. **b** Heatmap of the differentially expressed genes (i.e. $P < 0.05$) in chronic (1 week) flavopiridol-treated B16/F10 cells (log2 row normalized). **c** Expression change in TE$^{deff}$ vs. TE$^{prof}$ cancer tissues (left), and genomic lengths (middle), of repressed and over-expressed genes in flavopiridol-treated cells, right) Expression change of Class I and Class II genes in flavopiridol-treated cells. **d** Calculation of the traveling RNAP II ratio from total RNAP II ChIP-seq data, which is defined as the ratio of occupancy of RNAP II along the gene bodies to that around the TSS. **e** Genome-wide comparison of RNAP II traveling ratio in the control and chronic flavopiridol-treated B16F10 cells. Each point depicts a gene, and Class I and II genes are colored accordingly. **f** Log fold change in RNAP II occupancy in the promoter and gene body regions in chronic flavopiridol-treated cells over control in Class I and II genes. **g** Spearman's rank correlation of gene body RNAP II occupancy with overall polyA + mRNA expression (based on RNAseq data from **b**) in the respective genes in the parental and flavopiridol-treated B16F10 cells. **h** qPCR-based quantitation of expression of the indicated Jak1 exons in the parental and flavopiridol-treated B16F10 cells with and without 2 h stimulation with IFNγ. Error bars reflect s.d. of duplicate measurements. Statistical significance of difference is coded: (.): $P < 0.1$, (*): $P < 0.05$ by $T$-test. Boxplots: middle line: median, boxed areas extend from the first to third quartile; whiskers show 1.5x inter-quartile range from the first (bottom) or third (top) quartile

challenge significantly increased tumor cell seeding of control, and to a lesser extent, of flavopiridol pre-treated, B16/F10-OVA cells (Fig. 7d), showcasing the prominent role of NK-mediated regulation of cell seeding in this system. Importantly, seeding of control B16/F10-OVA cells was comparable to that of flavopiridol pre-treated cells in NK depleted mice, suggesting that flavopiridol pre-treatment rendered the B16/F10-OVA cells resistant to NK cell-mediated killing (Fig. 7d).

**TE$^{deff}$ induction protects from checkpoint inhibitor therapy.** Next, to test the causal role of transcription elongation defects in the resistance to immune checkpoint inhibitor therapy in vivo, we injected control and flavopiridol pre-treated CT26 colon carcinoma lines into immune-competent Balb/c mice, and measured tumor growth after treatment with the control IgG or anti-

CTLA4 antibodies. Anti-CTLA4 therapy significantly slowed the growth of control CT26 cells (Fig. 7e, f). However, CT26 cells pretreated with flavopiridol were not affected by anti-CTLA4 treatments (Fig. 7e, f). In line with the established role of IFN-γ/ antigen presentation pathway in the immune checkpoint inhibitor response[33–35], while tumors from the control CT26 line had high expression of H2-K$^D$ (HLA-I) and an induction of B2M upon anti-CTLA4 treatment, those from the flavopiridol pretreated line did not (Fig. 7g). Importantly, tumors from flavopiridol pre-treated cells retained TE$^{deff}$-like reduction in the total and phospho-RNAP II levels 3 weeks after flavopiridol release and in vivo growth (Supplementary Fig 12B), indicating relative stability of the phenotype in vivo.

Combination treatment with anti-CTLA4 and anti-PD1 therapy has shown significant improvement in the response and

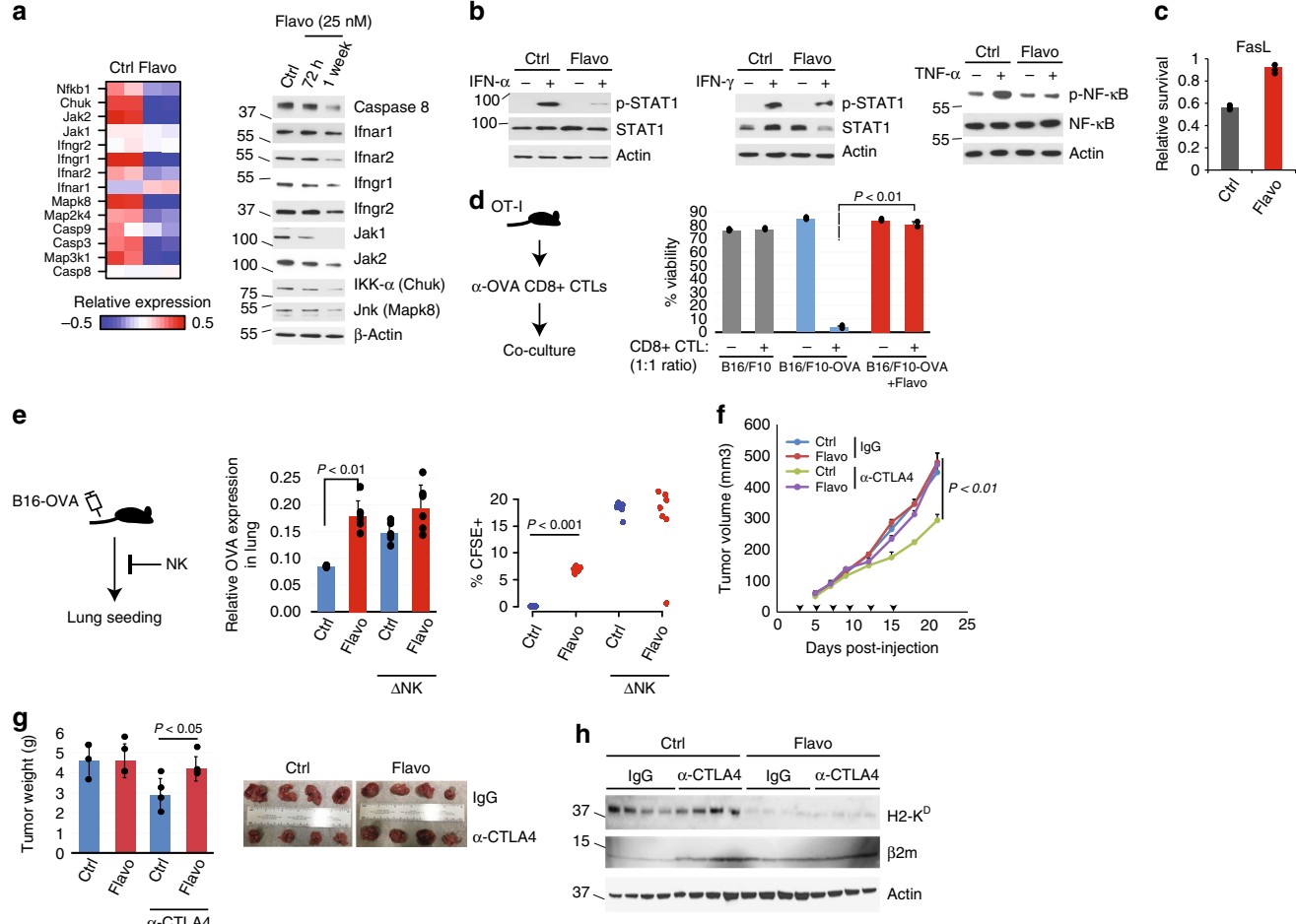

**Fig. 7** Chronic TE defects confer resistance to anti-tumor immune attack and immunotherapy in vitro and in vivo. **a** Heatmap of relative expression (left), and immunoblots for proteins (right), of some key inflammatory response pathway genes in control and flavopiridol-treated cells. **b** Control and flavo pre-treated cells were stimulated with IFN-α, IFNγ or TNF-α for 30 mins or FasL (**c**) for 24 h, and the readout was measured with the indicated proteins by immunoblotting (**b**) or viability assay (**c**). **d** Relative viability of B16/F10-OVA cells co-cultured with activated CD8 + CTLs (1:1 ratio) isolated from the spleens of OT-I mice. *P*: Welch two-sample *t*-test, **e** in vivo assessment of resistance to NK-mediated anti-tumor response: mice are injected intravenously with 2 × 10^5 B16/F10-OVA cells pre-stained with CFSE, and tumor cell seeding in the lungs is assessed 1 h later by measuring OVA mRNA levels in the lungs by qPCR (middle) or by flow cytometry for CFSE + cells (right). ΔNK: NK cell depletion by subcutaneous pre-injection of mice with anti-asialo GM antibody 1 day prior to tumor cell injection. The *p*-value is for Welch two sample *t*-test. **f** Tumor growth of control or flavopiridol pre-treated CT26 mouse carcinoma cells in syngeneic (Balb/c) mice treated with control (IgG) or anti-CTLA4 antibody (9H10), *P*: Welch two-sample *t*-test. **g** Weights and images (bottom) of excised tumors of indicated treatment groups after 21 days post-injection. **h** Immunoblots of some of the excised tumors from the treatment groups in experiment in **f** for H2-K^D (HLA-I), β2m, and Actin. Error bars: standard deviations of 3 (**c**), 2 (**d**), 6 (**e**), and 4 (**g**) replicates

overall survival rates in multiple cancers in the clinic. We tested if flavopiridol pre-treatment will also confer resistance to a combination of anti-CTLA4 and anti-PD1 therapy. Importantly, although the combination therapy showed a substantial efficacy in the control CT26 tumors, flavopiridol pre-treatment still conferred significant resistance, although at a lesser level than it did in anti-CTLA4 treatment (Supplementary Fig 12C, D). Therefore, chronic perturbation of the transcription elongation in TE^deff impairs the expression and function of the inflammatory response pathways, and confers resistance to anti-tumor immune attack and immune checkpoint blockade therapy (Fig. 8).

## Discussion

Transcriptional deregulations are frequently observed in cancers: splicing and intron retention defects[7,43], alternative poly-adenylation[2], and transcription read-throughs[5], and they contribute to different aspects of tumorigenesis. Here, we have identified a previously unknown phenotype in cancers that is

most prominently defined by defective transcription elongation of stimulus-responsive genes. This leads to significant repression of a panel of stimulus-responsive signaling pathways, among them those involved in pro-inflammatory response, at both mRNA and protein levels. Impaired response to IFN and cytolytic immune cell attack rendered the cells with TE^deff resistant to immunotherapy in vivo and in the clinic.

Tumor immunotherapy aims to boost the host anti-tumor immune responses by either direct immune-mediated attack of the tumor cells (adoptive cell therapy, monoclonal antibody therapy) or by removing the inhibitory checkpoint mechanisms (checkpoint inhibitor therapy). Immunotherapy is revolutionizing cancer care with a promise of cure for a select population of patients, and is rapidly being adopted as standard-of-care for multiple different cancer types[44]. Although many of the responding patients have durable complete remissions, the majority of patients do not respond to immunotherapy, and a large fraction of those who do eventually relapse[35].

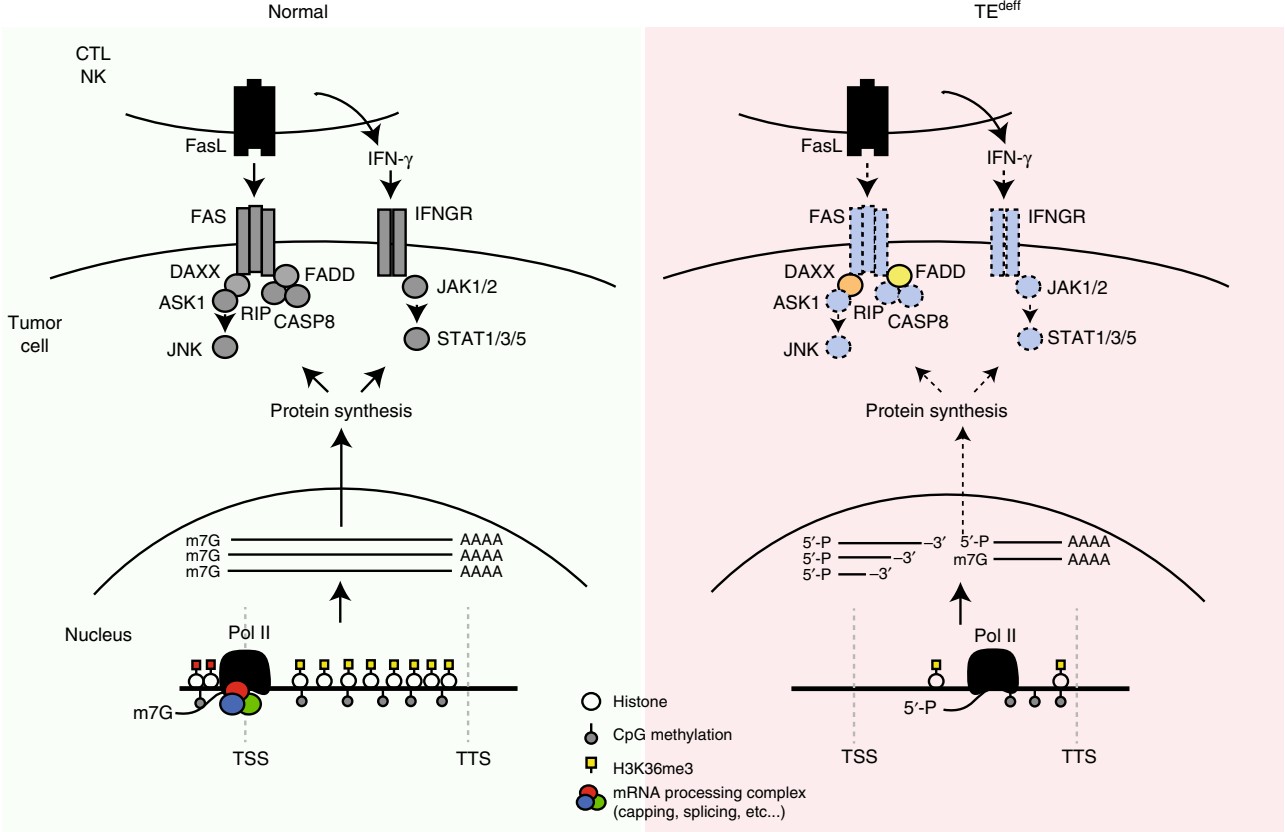

**Fig. 8** A model of TE$^{deff}$-induced immunotherapy resistance. Intact transcriptional fidelity is essential for the baseline and cytokine (e.g. IFN)-inducible expression of the inflammatory response pathways that prime the tumor cell for cytolytic cell attack (left). However, in TE$^{deff}$ cells, transcription elongation defects lead to the expression of spurious transcripts and poor nuclear export of the inflammatory response pathway genes. This, in turn, severely reduces the sensitivity of the tumor cell to the cytolytic T and NK cell-mediated anti-tumor mechanisms (right)

Unfortunately, the mechanisms of tumor resistance to these agents in the clinic are not fully understood.

Anti-tumor immunity is thought to be primarily driven by T-cell-mediated adaptive response to the neo-antigenic epitopes created by somatic mutations[45,25–28]. This type of immune surveillance involves the recognition by the T-cells of the neo-antigens presented by the tumor cells. As such, a common escape mechanism for tumor cells from immune surveillance is by mutations in the antigen presentation machinery, most notably in the *JAK/STAT/B2M* pathway, which are also enriched among patients with acquired resistance to immunotherapy[29,30,32]. Intriguingly, TE$^{deff}$ also leads to the inactivation of the innate inflammatory response pathways, including the IFN-γ/JAK/STAT and the antigen presentation pathways; however, in an epigenetic manner by impairing their mRNA expression (Figs. 3 and 5). Importantly, the expression of many key genes in these pathways are regulated at the level of transcription elongation, and as such, they are acutely sensitive to the perturbation of RNAP II elongation[10–12]. Our results show that chronic perturbation of RNAP II elongation in TE$^{deff}$ has a widespread effect on multiple aspects of mRNA expression, and has a specific inhibitory effect on the expression of the inflammatory response pathway genes (Fig. 7). As such, TE$^{deff}$ predicted poor clinical benefits in 4 independent cohorts, including both cytokine and immune checkpoint inhibitor immunotherapy (Fig. 4), and conferred resistance to immune checkpoint inhibitor treatment in vivo (Fig. 7). Importantly, TE$^{deff}$ tumors of kidney cancers and melanomas had higher infiltration by T-cells (judged by the expression of CD247) (see Supplementary Fig 9), suggesting that

the resistance to immune attack is probably at the level downstream of T-cell activation and infiltration, reminiscent of tumors progressing on immunotherapy[45]. Therefore, TE$^{deff}$ is a novel cancer phenotype that confers resistance to anti-tumor immune attack, and thus may be beneficial for proper stratification of immunotherapy-candidate patients, especially in the cases where alternative treatment options are available (e.g. ccRCC).

## Methods

**Cells and reagents**. UACC-812 and MDA-MB-415 cells were purchased from ATCC (Manassas, VA). RS4.11 and HL-60 cell lines were obtained from Ashish Kumar lab (CCHMC). All cell lines were authenticated by STR profiling at Genetica (Burlington, NC). UACC-812 cells were grown in Leibovitz's L-15 (Gibco) medium with 2 mM L-glutamine containing 20% fetal bovine serum (FBS) and 0.1% antibiotic and antimycotic (Gibco). MDA-MB-415 cells were grown in Leibovitz's L-15 (Gibco) medium with 2 mM L-glutamine supplemented with 10 µg/ml insulin (Sigma), 10 µg/ml glutathione (Calbiochem), 15% FBS and 0.1% antibiotic and antimycotic (Gibco). SKBR3, BT474, MDA-MB-231, CAL51, T47D cells were cultured in RPMI 1640 (Gibco) containing 10% FBS with 0.1% antibiotic and antimycotic (Gibco). MDA-MB-453 cells were cultured in improved minimum essential medium (Gibco) containing 20% FBS with 0.1% antibiotic and antimycotic (Gibco). All cells were cultured in a humidified atmosphere in 5% $CO_2$ at 37 °C. All cells were regularly tested for mycoplasma contamination and were negative.

**Datasets**. Processed RNAseq (gene-level, isoform-level, and exon-level expression), RPPA, somatic mutations, copy number variations, and Affymetrix exon array from The Cancer Genome Atlas (TCGA) were downloaded from the GDAC Firehose (https://gdac.broadinstitute.org/). Clinical survival data for patients in TCGA were obtained from Liu et al.[46]. The RNAseq BAM files were downloaded from the NCI Genomic Data Commons with authorization. The raw RNAseq data for the Van Allen and Hugo cohorts in Fig. 5 were downloaded from the dbGAP with authorization.

**Immunoblotting**. Total proteins were extracted with RIPA buffer (Santa Cruz Biotechnology, sc-24948), and 15 µg protein from each sample was run in a 4–18% SDS polyacrylamide gel (Bio-Rad), and transferred onto polyvinylidene difluoride membranes. The membranes were blocked in 5% dry milk in tris-buffered saline–Tween 20 for 1 h. Blocked membranes were incubated overnight with primary antibodies (see Supplementary Table 3) in 5% bovine serum albumin. After washing and incubating with the appropriate secondary antibody, protein signals were detected with enhanced chemiluminescence (Millipore). Original images of key blots in the manuscript are provided in the Supplementary Fig 14.

**PolyA tail mRNA capture**. Total RNA was extracted from the cells using Tri reagent (Sigma), followed by rRNA depletion and subsequent concentration of rRNA-depleted samples using RiboMinus™ Eukaryotic Kit (Ambion) according to manufacturer's instructions. PolyA + -RNA was isolated from rRNA-depleted samples using Dynabeads® Oligo(dT)25 (Ambion) according to the manufacturer's instructions. Purity and concentration of RNA yield were measured by NanoDrop (Thermo Scientific). The 260/280 ratio was 1.90–2.00, and the 260/230 ratio was 2.00–2.20 for all RNA Samples.

**5′ Capped RNA immunoprecipitation**. Five-prime capped RNAs were immunoprecipitated with the monoclonal 7-Methylguanosine antibody (BioVision) coated protein A columns, from total RNA devoid of rRNA using RiboMinus™ Eukaryote Kit (Ambion) according to manufacturer's instructions. Purity and concentration of RNA yield were measured by NanoDrop (Thermo Scientific). The 260/280 ratio was 1.90–2.00, and the 260/230 ratio was 2.00–2.20 for all RNA samples.

**Nuclear/cytoplasmic mRNA fractionation**. Qiagen RNeasy mini kit (Qiagen, Germantown, MD, USA) was used to extract total RNA from cells according to the manufacturer's instructions. Using total RNA as input, Cytoplasmic & Nuclear RNA Purification Kit (Norgen, Belmont, CA, USA) was used to isolate and purify cytoplasmic and nuclear RNA. Ribosomal RNA in the nuclear and cytoplasmic RNA fractions was depleted using RiboMinus Eukaryote Kit (Invitrogen, Carlsbad, CA, USA) according to the manufacturer's instructions. Ribo-depleted RNA was reverse transcribed to cDNA using RevertAid RT Kit (Thermo Scientific, Waltham, MA, USA). qRT-PCR was performed using GoTaq qPCR Kit (Promega, Madison, WI, USA) to detect genes expression levels. U-type snRNA primers were used to test the purity of nuclear and cytoplasmic fractions.

**Differential exon expression heatmap**. The t-statistic of difference (−log10 P-values in the case of cell line data, see Fig. 2c) in the expression of each exon (RPKM) was calculated between TE$^{deff}$ and TE$^{Prof}$ tumor samples. Every "expressed" gene (i.e. has a 90%-ile normalized count of >30 in a given cancer (e.g. KIRC) dataset) was defined by 20 exon bins (genes with <20 exons were stretched, and those >20 exons were compressed, into 20 bins), and corresponding exon t-values were visualized in a heatmap where columns (bins) were ordered from 5′ to 3′. For exon analyses, pre-computed RPKM values were used as provided in TCGA data matrix.

**Scoring TE$^{deff}$ in tumor samples**. Short and long isoforms of AT genes were identified as the mutually exclusively expressed clusters of isoforms (see Supplementary Fig 1c and legend). TE$^{deff}$ score (also TS score) was defined as the log2 ratio of average expression (normalized counts) of short to the average expression of long isoforms. Samples with log ratios >−1 were defined as TE$^{deff}$ (or TS+ in Fig. 1). See below for the definition of TE$^{deff}$ in total RNAseq samples from Van Allen et al.

**Intron retention analyses**. RNAseq reads are mapped using TopHat[47]. The bam files were then processed using custom python script using the pysam library to extract read counts of exon–exon junctions and exon–intron junctions. Briefly: for each gene, reads are extracted from the genomic regions defined by the start and stop site. Split reads with 8 bp anchors (a minimum of 8 bp mapped to each exon) and read mapping quality >20 are extracted and the junction is annotated by the start and stop positions of the gap. The number of reads mapping to each exon–exon junction is counted. For every exon–exon junction, identified reads ±150 bp around the exon–intron and intron–exon junctions are extracted, and the expression of these junctions is counted as the number of reads that span across the exon-intron/intron-exon junction with read mapping quality >20 and at least 8 bp on each corresponding exon and intron. For the ratio analyses of exon–intron and exon–exon junction reads, only exon–exon junction reads with at least 5 mapped reads and the intron length >500 bp were used. Using different cutoffs for either of these parameters did not significantly affect the results.

**ChIP-seq data analyses from roadmap epigenomics**. We examined different regions of the gene bodies of gene sets that are repressed (Class I genes in Fig. 2) or overexpressed (Class II genes in Fig. 1) in TE$^{deff}$ tumors for overlap with a large collection of genome-wide functional genomics datasets. We first compiled data relevant to gene regulation from a variety of sources including ENCODE[48],

Roadmap Epigenomics[49], the UCSC Genome Browser[50], and Pazar[51]. For both gene sets, we broke the constitutive genes into different regions, and overlapped these regions with each of the 2345 functional genomics datasets. We considered 3 regions in total: (−1000, +1) relative to the transcription start site (TSS) (promoter), all exons and all introns.

To illustrate, consider the promoter regions of the Class I gene set. For each gene in the set, we looked up the genomic coordinates of its promoter, and intersected these coordinates with each of the 2345 datasets. We calculated the observed overlap between the set of promoters and a given dataset as the number of promoters that overlap that dataset by at least one base. We then determined how significantly different the observed overlap was from the expected overlap with each dataset. To do so, we created a matched random set of promoters. For each gene in the Class I set, we randomly picked a gene from our background set of 10,448 expressed genes (from the heatmap in Fig. 1d), and generated a simulated promoter by matching the promoter length of the corresponding gene in the Class I set. This procedure therefore guarantees that the promoter length distribution of the random set will match the real set. For this random set, the overlap with each dataset was then calculated. This procedure was repeated 1000 times, resulting in a distribution of expected overlaps between the promoters and each dataset that follows a normal distribution, which we used to generate a Z-score and P-value for the observed number of overlaps. For example, if 50/100 promoters overlapped peaks from a given ChIP-seq dataset, and we expected 10±5, this yields a Z-score of 8. This procedure was repeated for each of the 3 gene regions listed above. To compare between the Class I and II gene sets, we calculated delta values based on the difference between the two Z-scores. This resulted in a list of genomic features specific to the gene regions of the Class I set relative to the Class II set, and vice versa.

**Survival analyses of TCGA datasets**. Patient stratification was done by classifying patients into non-exclusive lists based on drugs they received. Since drug annotations were not consistent (i.e. the same drug was annotated with different spellings and names for different patients), we compiled a vocabulary of immunotherapy drug annotations in the TCGA clinical samples for SKCM and KIRC. For immunotherapy drugs, our vocabulary included Alferon, GM-CSF, IL-18, IL-2, IL2, interferon, Interferon, Interferon-?2, Interferon-alfa, Interferon alfa, Interferon alfa-2b, interferon alpha, Interferon alpha, Interferon Alpha, Interleukin-2, Interleukin − 2, Laferon, Leukine, Alpha Interferon, IFN-Alpha (Intron), IL-2 (high dose), IL-2 Thearpy (interleukin), INF, interferon-alpha, interleukin-2, Interleukin 2-high dose, Intron A, Proleukin and proleukin (IL-2). For checkpoint inhibitor therapy, we considered ipilimumab, Yervoy, pembrolizumab, Pembrolizumab, and Ipilimumab annotations.

**Analyses of the hugo cohort**. RNAseq BAM files from the Hugo et al dataset were downloaded from GEO (GSE78220). Isoform level expression was estimated using kallisto[52], and intron retention was estimated as described above. The TE$^{deff}$ analyses were conducted by employing the pipeline in Supplementary Fig 1: as log ratio of average expression of short to full-length isoforms. TE$^{deff}$ population was defined as those that have >−1 log ratio of short/long isoform expression. Consistent with TCGA data, TE$^{deff}$ samples in this cohort also had higher retention of exon–intron junctions in class I genes (Supplementary Fig 13a).

**Analyses of the van allen cohort**. The RNA for sequencing in this cohort was obtained from formalin-fixed paraffin-embedded tissues[25]. Accordingly, the libraries were constructed by rRNA depletion rather than polyA selection. Since our pipeline established for TCGA (Supplementary Fig 1) and other (Supplementary Fig 5) tissue samples was for polyA-selected RNAseq, we used an alternative method of defining TE$^{deff}$ in this cohort. TE$^{deff}$ -like phenotype in this cohort was defined as increased global retention of exon-intron junction reads in Class I genes, as TE$^{deff}$ samples had almost exclusive increase in these values relative to other cases (Fig. 1i). TE$^{deff}$ and non-TE$^{deff}$ populations were defined by a cutoff at the median (21 samples each). The gene-level transcriptomic signature of TE$^{deff}$ samples defined this way highly correlated with the TE$^{deff}$ signature in TCGA for metastatic melanoma (Spearman's $\rho = 0.38$, $P < 10^{-60}$). Moreover, they had a significant enrichment of the exonic reads around the transcription start site, which decreased into the gene body, especially in the Class I genes, while those for the Class II genes were consistently upregulated (Supplementary Fig 13b, c). These observations are highly consistent with the exon array profiles of total mRNA from TE$^{deff}$ samples from TCGA (see Fig. 1e), which confirms that these samples are enriched for the TE$^{deff}$ phenotype. Choosing a different cutoff for TE$^{deff}$ samples instead of the median (e.g. upper 25%-ile) produced survival plots similar to those shown in Fig. 3d (see Supplementary Fig 13D).

To score TIL, we used the average expression of GZMK and PRF1 (perforin), and TIL-high and TIL-low populations were again determined by a cutoff at the median. Using just GZMK instead of the average, or just PRF1, or GZMA instead of GZMK, in lieu of TIL gave similar results.

**RNA sequencing of cell lines**. Total RNA was extracted from the cells using Tri reagent (Sigma). RNase-free DNase was used for removing all genomic DNA contamination. The RNA was precipitated by Isopropanol (Sigma), washed by ice

cold 75% ethanol (Sigma), and air dried prior to resuspension in 20 μl of DEPC-treated water. Purity and concentration of RNA was measured by NanoDrops (Thermo Scientific). The 260/280 ratio was 1.90–2.00 and the 260/230 ratio was 2.00–2.20 for all RNA samples.

Directional polyA RNA-seq was performed by the Genomics, Epigenomics, and Sequencing Core (GESC) at the University of Cincinnati. NEBNext Poly(A) mRNA Magnetic Isolation Module (New England BioLabs, Ipswich, MA) was used for polyA RNA purification with a total of 50 ng to 1 μg of good quality total RNA as input. The Core used Apollo 324 system (WaferGen, Fremont, CA) and ran PrepX PolyA script for automated ployA RNA isolation. NEBNext Ultra Directional RNA Library Prep Kit (New England BioLabs, Ipswich, MA) was used for library preparation, which used dUTP in cDNA synthesis to maintain strand specificity. In short, the isolated polyA RNA was Mg2+/heat fragmented (~200 bp), reverse transcribed to 1st strand cDNA, followed by 2nd strand cDNA synthesis labeled with dUTP. The purified cDNA was end repaired and dA tailed, and then ligated to adapter with a stem-loop structure. The dUTP-labeled 2nd strand cDNA was removed by USER enzyme to maintain strand specificity. After indexing via PCR (~12 cycles) enrichment, the amplified libraries together with library preparation negative control were cleaned up by AMPure XP beads for QC analysis. Libraries at the final concentration of 15.0 pM was clustered onto a single read (SR) flow cell using Illumina's TruSeq SR Cluster kit v3, and sequenced for 50 bp using TruSeq SBS kit on Illumina HiSeq system.

**RNA sequencing of RCC tissues**. RNA was isolated from 10–10 μm thick sections of OCT embedded kidney tumor tissue using a Qiagen RNeasy Micro Kit. It was then quantified and its integrity tested on an Agilent 2200 TapeStation. All of the samples had RNA Integrity Number (RIN) values >8.0. PolyA + RNA sequencing (2 × 100 bp at 50 M depth) was done at Beijing Genome Institute (BGI).

**ChIP-seq**. Chromatin immunoprecipitation was done largely as previously described[53]. For each ChIP sample, 3 μg antibody (RNAP II: Active Motif #39097, RNAP II – pSer5: Active Motif #61085) was coupled to 10 μl Protein G Dynabeads (Invitrogen # 10004D) following manufacturer's instructions. For each sample, one 10cm-plate of cells were cross-linked with 1.6% paraformaldehyde (Electron Microscopy Sciences #15710) in culture medium at room temperature for 40 min and then quenched by 135 mM glycine for 5 min. Cross-linked cells were washed twice with PBS, then lysed in 1 ml ChIP lysis buffer (50 mM HEPES pH7.5, 140 mM NaCl, 1 mM EDTA. 10% glycerol, 0.5% NP-40, 0.25% Triton X-100, 1x Protease Inhibitor Cocktail [Sigma #4693159001]) on ice for 1 h. Lysates were centrifuged at 5000 rpm for 5 min to pellet chromatin, then chromatin was resuspended in 1 ml Buffer 2 (10 mM Tris pH8.0, 200 mM NaCl, 1 mM EDTA, 0.5 mM EGTA, 1x Protease Inhibitor Cocktail), pelleted, then resuspended again in 1 ml Buffer 3 (same as Buffer 2 but without NaCl). Chromatin was sheared to 300 bp to 1 kb fragments using a Sonifier Cell Disruptor (Branson Ultrasonics #101-063-588) equipped with a microtip (Branson Ultrasonics #101-148-062) at 50% power output, intervals of 1-second on/1-second off, for a total of 7 min. Sarkosyl was added to a final concentration of 0.5% and the sheared chromatin was incubated for 10 min at room temperature, then centrifuged to remove debris and the supernatant chromatin was harvested. To 0.5 ml sheared chromatin, we added 150 μl ChIP mix (440 mM NaCl, 0.44% sodium deoxycholate, 4.4% Triton X-100) and antibody-coupled beads for immunoprecipitation overnight at 4 °C. ChIP samples were washed three times in Wash Buffer (50 mM HEPES pH7.5, 0.5 M LiCl, 1% NP-40, 0.7% sodium deoxycholate, 0.5 mM EDTA), then eluted in 50 mM Tris pH8, 1% SDS, 50 mM NaCl, 10 mM EDTA at 65 °C for 1 h. Both input and ChIP eluates were incubated at 65 °C overnight for reverse cross-linking. After sequential treatment with RNase and Proteinase K, DNA was purified by phenol chloroform extraction followed by precipitation in ethanol. Precipitated DNA fragments were processed to ChIP-seq library using the ThruPLEX DNA-seq 12 S kit (Takara #R400428) according to manufacturer's instructions. Libraries were submitted to CCHMC DNA Sequencing Core for Nextgen sequencing using Illumina HiSeq2500 with single-read, 75 bases, 10 million reads per sample.

**ChIP-seq data processing**. Chipseq reads were aligned to the reference genome (hg19 in human and mm10 in the case of mouse) using BWA-MEM[54]. Samtools is then used to exclude reads that were either unaligned, were not primary alignments, failed vendor QC, are PCR duplicates or have quality <10 (option: -F 1084). In the case of paired end sequencing (for mouse ChIP-seq for flavopiridol-treated cells) an additional filter is used to retain only properly paired reads (option: -f 2). The BAM files post filtering are indexed and coverage around genebody, TSS and TES and 2000bp flanking regions is computed using ngs.plot[55]. Briefly, reads from the BAM file are overlaid with regions of interest and coverage at single base resolution and the vector is normalized by length. Coverage vector is then fit to a spline and then 101 points are sampled at equal intervals and normalized by library size (per million reads).

**RNAseq data processing**. The fastq files from RNA-seq data (Fig. 2, Supplementary Fig 5, Fig. 6) were used to quantify gene and isoform-level expression using RSEM[56].

**Cytokine treatments**. Equal numbers of cells ($10^5$) were seeded into 12 well culture plates in their corresponding growth medium. Next day, cells were treated with IFN-γ, IFN-α (5 ng/ml), or TNF-α (5 ng/ml) for 45 min and protein was extracted in RIPA buffer.

**Cytotoxicity assay with FasL**. Equal number of cells was seeded into the wells of 96-well culture plates in their corresponding medium and incubated overnight in a 5% $CO_2$ humidified incubator. Cells were then treated with different concentrations of $h_{his6}FasL$ (0.1 ng/ml–1000 ng/ml) (Cell Signaling #5452) in the presence of 10 μg/ml anti-His antibody (Cell Signaling #2366 P) for 24 h. Dead cells were removed by washing with PBS buffer and the attached cells were fixed and stained with crystal violet solution [20% methanol, 0.5% crystal violet (Sigma) in 1 × phosphate-buffered saline (PBS)] for 30 min. Excess stain was removed by gently rinsing the plates in tap water, and the plates were dried at room temperature. Crystal violet crystals were redissolved in Triton (Amresco), and cell density was determined by measuring the absorbance at 570 nm in a microplate reader (Bio-Tek Instruments).

**Testing NK-mediated tumor cell killing in vivo**. Control and flavopiridol pre-treated (25 nM for 7 days) 2 × $10^5$ B16-OVA cells were injected into the tail veins of C57Bl/6 mice (8–10 weeks old, female). One hour later, the lungs were harvested, digested in liberase, and the frequency of tumor cells was assessed using quantitative PCR[42]. The mRNA levels for OVA (B16-OVA) were assessed and normalized to GAPDH. To demonstrate that the observed effect is NK cell dependent, parallel groups were treated with NK depleting agent anti-asialo GM1 (20ul, 24 h before the start of the experiment) (Wako Chemicals # 986–10001). 6 mice for each group were used. The protocol and use of mice were performed with the approval of the Cincinnati Children's Institutional Animal Care and Use Committee.

**OT-I CD8+CTL isolation and activation**. CD8+ cells were purified from spleens of OT-I mice by MojoSort™ Mouse CD8 T Cell Isolation Kit as per manufacturer's protocol. Purified OT-I cells were primed in vitro using a system consisting of an adherent fibroblast APC (MEC.B7.SigOVA) engineered to express a specific OVA-derived, H-2Kb-restricted peptide epitope OVA$_{257–264}$ (SIINFEKL), along with the co-stimulatory molecule B7.1. The adherent fibroblast APC were seeded at 75,000 cells per well in 24-well plates. After 24 h, the monolayer of APC was washed once with medium, and naive OT-I cells (0.5 × $10^6$) were added in 2 mL of IMDM supplemented with 50 mM β-ME, 2 mL EDTA, 4 mM L-Glutamine and HEPES and 10%FBS. After co-culture for 20 h, the non-adherent OT-I cells were gently removed and transferred for co-incubation.

**Co-culture of CD8+cells with B16/F10-OVA cells**. Upon harvesting from the fibroblast line, OT-I-derived CD8+ cells were co-cultured with B16/F10, untreated B16/F10-OVA, and B16/F10-OVA cells pre-treated with flavopiridol (25 nM) for 1 week, at a ratio of 1:1 (300,00 cells each) in complete DMEM media for 20 h. Cells were then stained by incubating them in cold PBS containing 0.5% FBS and 0.05% sodium azide with viability dye and relevant labeled antibodies. Fixable Live/Dead staining dye e780 was from eBioscience, AF647-conjugated mouse CD8 and BV421-conjugated mouse CD45 were from Biolegend. Viability of B16/F10-OVA and CD8+T cells was analyzed by BD FACSCanto (Becton Dickinson, San Jose, CA) and BD FACS Diva software.

**Tumor growth assay with anti-CTLA4 and –PD1 treatment**. Balb/C mice (6–8 weeks old, Jackson Laboratories) were injected with control or flavopiridol pre-treated (25 nM for 7 days) 1 × $10^6$ CT26 cells subcutaneously to the right flank. One-half of the mice were administered with 100 μg anti-CTLA-4 (clone 9H10) alone (for experiment in Fig. 7e), or together with anti-PD1 (RMP1-14; BioX Cell) (for experiment in Supplementary Fig 12C, D) antibodies in PBS, and the other half were given control IgG in PBS intraperitoneally on days 3, 5, 7, 9, 12, and 15 post implantation. Tumor volumes were monitored every two days up to day 21. The mice were euthanized, followed by tumor excision. Animal research was approved and overseen by The CCHMC Institutional Animal Care and Use Committee (CCHMC IACUC).

**Code availability**. Codes generated for the computational analyses in this study are available from the authors upon request.

## Data availability

All of the RNA- and ChIP-sequencing data have been deposited to GEO with accession number GSE119679 (super-series).

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

## Acknowledgements

This work was in part supported by NCI (CA193549) and CCHMC Research Innovation Pilot awards to KK, and Department of Defense (BC150484) award to NS. The genomic sequencing core at the University of Cincinnati is supported by P30-ES006096.

## Author contributions

V.M., N.S., E.C., B.M., L.P.V., and J.S.P. performed the experiments. V.M., K.C., X.C., K.C., N.S., M.T.W., E.M.J., and K.K. performed data analyses. N.R., S.W., G.H., and E.M.J. provided materials. K.K. directed the project and wrote the manuscript.

## Additional information

**Competing interests:** The authors declare no competing interests

