## [Peer Review File · Nature Communications]

Reviewers' Comments:

Reviewer #1:

Remarks to the Author:

Modur et al. present here a novel observation that defective elongation of transcription has direct implications on the anti-tumor immune response as well as response towards checkpoint blockade therapy. The Authors nicely show that some cancers have a defect in RNA elongation and thus have an altered transcriptome. Most prominently gene involved in innate immune signaling are affected an observation that correlates with reduced anti-tumor immunity.

Although the data are very compelling several point need to be clarified or improved.

Figure 1C if TS correlated with response to checkpoint blockade therapy then why is this not reflected here, meaning cancers with hi TS should not respond but kidney cancer shows responses Figure 3A it seems puzzling that only 4 cell lines have this phenotype, based on percentage found in cancer patients. What are potential reasons.

Figure 4A The graphs for TEdeff/prof and immunotherapy seem to be slightly different between the plots - why. The authors make the argument that TEdeff reduced sensitivity to immunotherapy but based on the presented data it more so indicates an increase in sensitivity to targeted therapy.

General comment are TEdeff cells only "stressed" tumor cells?

Figure 6 the authors nicely show decrease in immune mediated surveillance in Flavo treated cells but this induced more then one change -> it would be key to prove necessity and sufficiency of the proposed mechanisms to make the conclusion drawn here. Overexpression of FAS or IFNGR, insensitive to the shorter elongation would need to be performed - otherwise the conclusions are only correlations.

Reviewer #2:

Remarks to the Author:

In this manuscript, Modur and colleagues mine publically available mRNA-seq data from the TCGA and identified a subset of tumors with apparent defects in transcription elongation (which they refer to as TEdeff). The authors claim that these defects converge on stimulus-responsive, long genes, such as inflammatory pathway genes. This suppression of these genes caused an impaired response in inflammatory death stimuli and could be correlated to clinical resistance to immunotherapy. This leads the authors to the conclusion that global defects in transcription elongation represent a "unknown epigenetic mechanism of silencing of key inflammatory response pathways".

While potentially of interest, I am not convinced that conclusions drawn by the authors are supported by the (largely correlative) data. In detail, measurements of transcription elongation in experimental settings are not convincing:

- CDK9 inhibition does not mimic the phenotype of class I genes (lower expression of gene-body exons while increasing expression of later exons) but simply stops Pol II in the promoter-proximal pausing site - hence not leading to any poly-adenylated transcripts (of genes controlled by pause release - which is-based on the most recent estimates- around 80-90%).
- The duration/timing of these experiments is very questionable (see detailed comments below)

In general, I am not convinced that the phenotypes extracted by the authors from the TCGA data are really reflective of an elongation defect. To me, it rather seems that splicing defects would be an equally likely explanation

Many of the analysis are very cumbersome to understand and the manuscript in its current form is by no means an intuitive read. This is partly due to the way how the authors present the data and the scores they are using (e.g. FIRMA score).

In general, while some of the reported clinical phenotypes are pretty stunning, the mechanistic underpinnings of the apparent elongation defect are not worked out sufficiently. Hence, I don't recommend publication of the manuscript in the current form.

Specific remarks grouped by the respective Figure:

Figure 1 (and corresponding SI)

1A: the way the data is displayed is confusing to me. It seems that some information is encoded in the positioning along the x-axis, however that isn't clear. The boxplot depiction as in Supplementary 1E is much more intuitive (but would require reporting the statistics of the observed differences).

1E/F: is there a need to present the data as FIRMA scores, or is there a more intuitive way of displaying the data? In its current form, I am not convinced that this analysis allows the drawn conclusions. Especially related to 1F: Pol II pausing typically occurs 20-80 bp downstream of the TSS and shows an according graphs typically display a very sharp drop after the 150 bp after the TSS. This is not in line with the data here where a drop to baseline levels is only observed at 1kb downstream of the TSS.

1H: authors should show the entire gene. Display of the gene structure doesn't allow to see if this is the 3'UTR or just the last 3 exons. Any sort of scaling on the y-axis is also missing.

Figure 2:

Overall issue: Authors cite relatively old literature (Adelman 2009) to make the point that inducible genes are specifically controlled by pause release. More recent data (e.g. Mayer et al. 2015) indicates that around 90% of all human genes feature promoter-proximally stalled Pol II and are regulated by pol ii pausing.

2C: authors need to plot the enrichment characteristics (p-value, q-value, odds-ratio)

2D: this data is not "highly consistent" and to the best of my abilities, I can't recognize a correlation

Figure 3:

Doesn't this data suggest that the vast majority of all cell line models do not display a corresponding TEdef phenotype? The authors identified 4 out of how many in total?

3E based on these plots, the major difference is expression of Pol II itself. If the authors want to make the claim that elongation is abrogated in those lines, they need to show that ser2-pol II/ total Pol II ratio is shifted. Ideally, authors would perform total Pol II ChIP-seq in all 4 cell lines and compare pausing indices.

Minor remark: the authors compare 2 AML cell lines with a CML cell line (K562) and a B-ALL cell line (KOPN8). It would be more telling to compare it to other AML cell lines (MLL-rearranged and non MLLr)

Figure 4:

4F,G: what comparisons are the reported p-values referring to?

Figure 5:

5B: none of these measurements are directly indicating an elongation defect, but only show processing defects. Pol II (or Ser-2 Pol II) ChIP-qPCR (or ChIP-seq) would be required to make that point

5C: only a handful of those proteins seem significantly changed. Looking at the breast cancer cells: no change in IFNAR1, IFNAR2, IFNGR1, IFNGR2

Figure 6:

A: treatment durations are very worrisome. Ser2-P levels should be affected as early as 1h post treatment. Measurements at 3 days or a week are by definition prone to interpreting indirect and downstream effects

B: timing issues as in A. concentrations of flavopiridol that wipe out ser2p (as shown in 6A) lead to global defect in mRNA levels (meaning almost all mRNAs should be downregulated) and it is very doubtful that one would see relatively balanced effect in magnitude of up-and down regulated genes. Without spike-in controls, these measurements are inconclusive.

F,G: the authors should comment on why they did pre-treatment experiments. Reversal of kinase

inhibitor effect (and hence re-expression of hypersensitive genes) should occur relatively quickly. How do the authors explain that the see effect duration up to several weeks?

Reviewer #3:

Remarks to the Author:

This is a compressive study to describe the novel link between defective transcription elongation in cancers and treatment response. However, there are some questions the authors need to clarify and address:

- 1) For TCGA data, the authors did not provide sufficient information how isoform and gene level data were obtained (directly from TCGA level III?). About 10 cancer types were analyzed. What happened to other cancer types and how the authors decided to keep these in the paper? The number of samples used for each is also not clear.
- 2) The authors evaluated processing methods, sample quality or batch for potential artifact but they were limited (to KIRC). What about mapping methods? Would tumor stroma affect it, particularly from immune cells?
- 3) How many cell lines from CCLE were analyzed? The defect event proportion appears much lower than the solid tumors (20%). How is this explained?
- 4) Fig G: how were the TE_{deff} vs. TE_{prof} overall expression levels like? Did the authors normalize that for the plot?
- 5) The authors mentioned mutations did not affect the finding but did they look at indel mutations (micro or short indels)? How about SNVs? Recent studies did report indel burden [PMID: 28694034] or type I interferon deletion [PMID: 29618619] affected immune treatment response.
- 6) I did not see the gene list with defective transcription elongation which should be provided in supplementary material.
- 7) Reference for "kallisto" should be provided

Response to Reviewers:

Reviewer #1 (Remarks to the Author):

Modur et al. present here a novel observation that defective elongation of transcription has direct implications on the anti-tumor immune response as well as response towards checkpoint blockade therapy. The Authors nicely show that some cancers have a defect in RNA elongation and thus have an altered transcriptome. Most prominently gene involved in innate immune signaling are affected an observation that correlates with reduced anti-tumor immunity.

Although the data are very compelling several point need to be clarified or improved.

Figure 1C if TS correlated with response to checkpoint blockade therapy then why is this not reflected here, meaning cancers with hi TS should not respond but kidney cancer shows responses

Response: Although renal cancers are abundant in the TEdeff phenotype, they are also highly infiltrated by T-cells, which is why they show relatively high responses to cytokine immunotherapy (IL2, IFN) and immune checkpoint blockade. We showed that the non-responders are enriched for the TEdeff phenotype. Given that the rate of response to immunotherapy in kidney cancers or melanomas in the range of ~20-40%, it is not surprising that there is no direct correlation of TEdeff rates in cancers with their overall response rates to ICB.

Figure 3A it seems puzzling that only 4 cell lines have this phenotype, based on percentage found in cancer patients. What are potential reasons.

Response: We agree that this is puzzling. We only detected a few cell lines out of ~1,000 in CCLE, and validated 4. One of the reasons might be that the TEdeff phenotype may not be stable in vitro, or that TEdeff+ cells do not grow well in vitro. The former is supported by our observation that the TEdeff-like phenotype caused by chronic flavopiridol treatment is highly sustainable in vivo in the absence of flavopiridol (see revised Supp.Fig.12b), whereas it is lost faster in vitro for the same cells (see below).

Figure 1. CT26 cells were treated with 25nM flavopiridol for 7 days, and released from flavopiridol for the indicated number of days.

Figure 4A The graphs for TEdeff/prof and immunotherapy seem to be slightly different between the plots - why.

Response: There are a few patients who were assigned both therapies (1 patient among TEdeff, and 7 patients among TEprof). We assigned them to one or the other in the bottom plots in Fig.4A, which creates small differences between the top and bottom plots. However, excluding these patients altogether does not change the pattern or the statistics significantly.

The authors make the argument that TEdeff reduced sensitivity to immunotherapy but based on the presented data it more so indicates an increase in sensitivity to targeted therapy.

Response: There was no significant difference in the overall survival of TEdeff and TEprof patients treated with targeted therapy (see below).

Figure 2. KM plot of TEdeff (red) and TEprof (blue) KIRC patients treated with targeted therapy ($P = 0.49$).

suggesting that TEdeff tumors are not more responsive to targeted therapy relative to TEprof tumors. It rather seems that they are more resistant to immunotherapies.

General comment are TEdeff cells only "stressed" tumor cells?

Response: TEdeff tumors have overexpression of some metabolic and proteostatic pathways (not shown), so it is possible that they are under certain types of stress, especially considering the severely perturbed RNA homeostasis. We are considering the possibility that certain types of stress may lead to a TEdeff-like phenotype. For example, nucleotide depletion in yeast has been shown to lead to transcription elongation defects, suggesting that a similar environmental condition may lead to the TEdeff phenotype in tumors. However, we do not believe that the TEdeff phenotype we are observing in clinical samples is a product of generic stress response. Rather, it is more likely that TEdeff creates certain types of metabolic and proteotoxic stress due to aberrant mRNA synthesis. The latter is an area of active investigation in our lab currently.

Figure 6 the authors nicely show decrease in immune mediated surveillance in Flavo treated cells but this induced more then one change -> it would be key to prove necessity and sufficiency of the proposed mechanisms to make the conclusion drawn here. Overexpression of FAS or IFNGR, insensitive to the shorter elongation would need to be performed - otherwise the conclusions are only correlations.

Response: We now show that overexpressing Jak1 is sufficient to reverse many of immune response signaling defects in flavo-treated cells. Jak1 overexpression resensitized TEdeff CT26 cells to IFN γ and FasL treatments, and prevented the suppression of several key immune response pathway genes (Supp.Fig.11c-e). These observations establish a causal role for the loss in the expression of key Jak1/Stat pathway genes in TEdeff-mediated immunotherapy resistance.

Reviewer #2 (Remarks to the Author):

In this manuscript, Modur and colleagues mine publically available mRNA-seq data from the TCGA and identified a subset of tumors with apparent defects in transcription elongation (which they refer to as TEdeff). The authors claim that these defects converge on stimulus-responsive, long genes, such as inflammatory pathway genes. This suppression of these genes caused an impaired response in inflammatory death stimuli and could be correlated to clinical resistance to immunotherapy. This leads the authors to the conclusion that global defects in transcription elongation represent a "unknown epigenetic mechanism of silencing of key inflammatory response pathways".

While potentially of interest, I am not convinced that conclusions drawn by the authors are supported by the (largely correlative) data. In detail, measurements of transcription elongation in experimental settings are not convincing:

- CDK9 inhibition does not mimic the phenotype of class I genes (lower expression of gene-body exons while increasing expression of later exons) but simply stops Pol II in the promoter-proximal pausing site - hence not leading to any poly-adenylated transcripts (of genes controlled by pause release - which is based on the most recent estimates- around 80-90%).

Response: It is true that acute CDK9 inhibition with very high doses of flavopiridol (or DRB) (~200nM) will halt Pol II at the promoter-proximal sites in most genes. This level of Pol II inhibition is obviously not viable, nor is it what we observe in TEdeff tumors. However, the role of TE factors in repressing internal promoter usage is well-established in the field, and TE inhibition leads to extensive transcription from within cryptic internal promoters (see Kaplan et al Science 2003, Chu et al EMBO J 2007, Venkatesh et al Nature 2012). Therefore, acute high-dose inhibition of CDK9 may have a substantially different profile from chronic low-dose inhibition.

We are using very low dose of flavopiridol (25nM), ten times less than that used in the studies referred to by the reviewer, and for a prolonged time (1 week), precisely to mimic chronic, viable, low-level defects of Pol II elongation in TEdeff tumors. We showed using RNAseq, that Class I genes are disproportionately affected by this treatment (Fig.6). In addition, we now also performed Pol II ChIP-seq in these cells, and not only show again that Class I genes are disproportionately affected, but also that chronic low-dose CDK9 inhibition can lead to spurious non-productive transcription. Moreover, we have done detailed exon-level qPCR analysis of Jak1 expression in control and flavopiridol-treated (low dose 1 week) cells, and show strong evidence of increased expression of internal exons in the flavopiridol-treated cells (Fig.6). Therefore, chronic sub-lethal TE inhibition may lead to the array of defects, including cryptic intragenic promoter usage, in TEdeff tumors.

- The duration/timing of these experiments is very questionable (see detailed comments below)

In general, I am not convinced that the phenotypes extracted by the authors from the TCGA data are

really reflective of an elongation defect. To me, it rather seems that splicing defects would be an equally likely explanation

Response: The TEdeff phenotype is a complex phenotype, and is characterized by not only transcription elongation defects, but also of splicing (see Fig.1H-I), mRNA capping and polyadenylation (Fig.3), nuclear export (Fig.3), DNA methylation (Fig.1G) and histone remodeling (Fig.3). It is important to note that these are in fact highly co-dependent processes, and the perturbation of TE has been shown to lead to others in multiple contexts (see review by Jonkers and Lis, Nat Rev Mol Cell Biol 2015). Our data with chronic flavopiridol treatment presented in Fig.6 also shows that.

Many of the analysis are very cumbersome to understand and the manuscript in its current form is by no means an intuitive read. This is partly due to the way how the authors present the data and the scores they are using (e.g. FIRMA score).

Response: we apologize for our unintuitive presentation of data and analyses. We have revised the manuscript text, figures and figure legends to make it more intuitive and easier to understand (see below for details).

In general, while some of the reported clinical phenotypes are pretty stunning, the mechanistic underpinnings of the apparent elongation defect are not worked out sufficiently. Hence, I don't recommend publication of the manuscript in the current form.

Specific remarks grouped by the respective Figure:

Figure 1 (and corresponding SI)

1A: the way the data is displayed is confusing to me. It seems that some information is encoded in the positioning along the x-axis, however that isn't clear. The boxplot depiction as in Supplementary 1E is much more intuitive (but would require reporting the statistics of the observed differences).

Response: The columns in heatmaps in Fig.1A are sorted based on short/long isoform ratio. We have now clearly indicated this in the revised figure.

1E/F: is there a need to present the data as FIRMA scores, or is there a more intuitive way of displaying the data?

Response: FIRMA scores reflect specific inclusion/exclusion events for exons controlling for overall gene expression. Therefore, this is an important metric for showing gene position-specific differential transcription. We have revised the figure legend to indicate that FIRMA scores reflect specific exon inclusion and exclusion events.

In its current form, I am not convinced that this analysis allows the drawn conclusions. Especially related to 1F: Pol II pausing typically occurs 20-80 bp downstream of the TSS and shows and according graphs typically display a very sharp drop after the 150 bp after the TSS. This is not in line with the data here where a drop to baseline levels is only observed at 1kb downstream of the TSS.

Response: As with the response to previous comments above, the effect described by the reviewer (Pol II pausing at 20-80bp downstream of TSS) is seen with acute high-dose inhibition of Pol II elongation (DRB or flavopiridol). Our Pol II ChIP-seq data show that chronic low-dose CDK9 inhibition does not lead to such a sharp peak at the TSS-proximal site. See this figure:

although there is a slight shift in the peak to the right. In addition, ChIP-seq with total Pol II presented in Fig.3 and Supp.Fig.7 clearly shows higher occupancy around the promoter, and lower occupancy within gene body for Class I genes, strongly suggesting elongation defects into the gene body. Therefore, chronic and sublethal elongation defects may have a substantially different profile from acute high-dose TE blockage.

1H: authors should show the entire gene. Display of the gene structure doesn't allow to see if this is the 3'UTR or just the last 3 exons. Any sort of scaling on the y-axis is also missing.

Response: We only showed the 3'-most part of the gene to highlight the exon definition/intron retention defects in the 3' exons of this gene. We have now included the entire gene in the revised Supp.Fig.4a. We have also included the scales on the y-axis.

Figure 2:

Overall issue: Authors cite relatively old literature (Adelman 2009) to make the point that inducible genes are specifically controlled by pause release. More recent data (e.g. Mayer et al. 2015) indicates that around 90% of all human genes feature promoter-proximally stalled Pol II and are regulated by pol ii pausing.

Response: The study by Mayer et al 2015 using NET-seq, and others using ChIP-seq, showed that ~90% of genes feature a higher Pol II or transcript density at the TSS (inverse traveling ratio ≥ 2). However, a promoter-proximal Pol II peak could have multiple interpretations (e.g. slower elongation or higher initiation rates), and does not indicate regulation at the level of pause-release (see Ehrensberger et al, Cell 2013). There is even a notion that a large fraction of "paused" Pol II in fact reflects rapid initiation-termination cycles by Pol II at certain genes (Krebs et al, Mol Cell 2017). Therefore, a gene featuring a high promoter-proximal Pol II density or activity is not necessarily regulated at the level of pause-release.

Multiple studies, including Mayer et al, have shown that genes have highly variable promoter Pol II density distributions and half-lives, as well as transcription initiation, pausing time and elongation rates. Therefore, naturally, there is a wide range of sensitivities to Pol II elongation inhibition. While very high doses of Pol II elongation inhibition (with DRB or flavopiridol) will inhibit most gene expression in the cell, we showed that low-level inhibition will have an effect that is biased towards the genes that are also repressed in TEdeff tumors (class I) (Fig.6).

2C: authors need to plot the enrichment characteristics (p-value, q-value, odds-ratio)

Response: All of the pathways depicted in this plot are enriched ($P < 0.001$, hypergeometric test) in the Class I (blue) and Class II (red) genes, respectively. The purpose of this plot is to show the differences in the distribution of gene lengths among these pathways. We have now indicated in the revised figure legend that all of the depicted pathways are enriched in the Class I and II genes, respectively.

2D: this data is not "highly consistent" and to the best of my abilities, I can't recognize a correlation

Response: Correlation below $r = 0.5$ are hardly visually discernible for untrained eyes, especially in large sample sizes. In fact, one would rarely expect obvious visually discernible correlations in such large samples unless there is an underlying artifact. A correlation of $\rho = 0.38$ for this sample size ($n \sim 160$ protein-mRNA pairs) is in fact highly significant, as indicated by the p -value. However, to aid in the visual interpretation, we now include boxed distributions of the data points in equal intervals overlaid on the original scatter plot, where a clear linear trend is visible.

Figure 3:

Doesn't this data suggest that the vast majority of all cell line models do not display a corresponding TEdeff phenotype? The authors identified 4 out of how many in total?

Response: Yes, it does. Please refer to our response to a similar question raised by Reviewer 3 below.

3E based on these plots, the major difference is expression of Pol II itself. If the authors want to make the claim that elongation is abrogated in those lines, they need to show that ser2-pol II/ total Pol II ratio is shifted. Ideally, authors would perform total Pol II ChIP-seq in all 4 cell lines and compare pausing indices.

Response: Chronic TE defects have been shown to lead to reduced total Pol II levels due to ubiquitination-mediated degradation (Somesh et al, Cell 2005); also see Chen et al Blood 2005 for chronic flavopiridol-induced decrease in Pol II levels. Our data with chronic flavopiridol and CCNT1-KO directly show that stable and chronic TE may lead to dramatically reduced total Pol II levels (Supp.Fig.11), perhaps due to the degradation of stably paused Pol II. We have performed total RNAP II chip-seq in a TEdeff (UACC-812) and a TEprof (T47D) cell line. In addition, to analyze aberrant transcription initiation events (e.g. intragenic), we also did RNAP II – Ser5p chip-seq in these cells. We not only show increased promoter occupancy of total Pol II (indicating defective TE) in TEdeff cells, but also increased occupancy at the TES-proximal sites of both total and Ser5-p Pol II, indicating spurious intragenic transcription initiation (Fig.3E-F). We are also providing Pol II chip-seq data for chronic flavopiridol-treated cells showing peculiar Pol II density profiles along gene bodies in chronic TE defects (Fig.6).

Minor remark: the authors compare 2 AML cell lines with a CML cell line (K562) and a B-ALL cell line (KOPN8). It would be more telling to compare it to other AML cell lines (MLL-rearranged and non MLLr)

Response: RS4.11 (TEdeff) and KOPN8 (TEprof) are B-ALL lines, HL-60 is AML and K562 is a CML line. We have now added two more AML lines: THP1 and Molm13 to the blots in Fig.3E.

Figure 4:

4F,G: what comparisons are the reported p -values referring to?

Response: The survival analyses were performed using COX proportional hazards model, and the p -values are by Wald test. This is now indicated in the figure legend.

Figure 5:

5B: none of these measurements are directly indicating an elongation defect, but only show processing defects. Pol II (or Ser-2 Pol II) CHIP-qPCR (or CHIP-seq) would be required to make that point

Response: Yes, these measurements were not intended as evidence of elongation defects, but as consequences of such. Elongation defects are known to cause mRNA capping, polyA-tion and nuclear export defects, as we also have shown with flavopiridol experiments in Fig.6 and Supp.Fig.11a.

5C: only a handful of those proteins seem significantly changed. Looking at the breast cancer cells: no change in IFNAR1, IFNAR2, IFNGR1, IFNGR2

Response: Yes, the suppression of the inflammatory response pathway genes is not over-the-board, but of several key proteins, which has been shown to be sufficient to confer resistance to immune attacks in both clinical settings (e.g. Shin et al 2016, Zaretsky et al 2016) and animal models (e.g. Patel et al 2017, Manguso et al 2017). We show that re-expressing one of these genes, Jak1, can reverse the resistance of flavopiridol-treated cells to IFN and FasL (Supp.Fig.11c-e).

Figure 6:

A: treatment durations are very worrisome. Ser2-P levels should be affected as early as 1h post treatment. Measurements at 3 days or a week are by definition prone to interpreting indirect and downstream effects

Response: This concern of the reviewer, like in the several points raised above, perhaps is due to equating (or confusing) TEdeff with an acute high-dose inhibition of Pol II elongation. We think it would be helpful to make distinctions between the two phenomena. Acute inhibition of Pol II elongation by flavopiridol or DRB in the past studies has been done to study the mechanisms of Pol II elongation. However, this level of inhibition, although helpful for studying acute phases of total inhibition of Pol II elongation, is not physiological, and no cell will survive this. It is very important to note that TEdeff tumors and cells are viable, and obviously, are still functionally expressing most of the transcriptome. Therefore, the level of transcriptional defects in TEdeff cells is substantially lower than the high-dose acute inhibition studies referred to by the reviewer.

Moreover, TEdeff is a chronic phenotype. As such, it is characterized with a plethora of additional downstream effects, such as defects in mRNA processing, splicing and nuclear export, along with defects in proteostasis (unpublished observations). Therefore, measuring the indirect and downstream effects of chronic Pol II elongation defect is in fact the intended purpose of this particular approach.

B: timing issues as in A. concentrations of flavopiridol that wipe out ser2p (as shown in 6A) lead to global defect in mRNA levels (meaning almost all mRNAs should be downregulated) and it is very doubtful that one would see relatively balanced effect in magnitude of up-and down regulated genes. Without spike-in controls, these measurements are inconclusive.

Response: The “wipe-out” effect is an issue with exposure. We are now showing a darker exposure of the same samples (Fig.6A). High-dose flavopiridol treatment will definitely lead to all mRNAs to be downregulated, which will cause cell death. However, 25nM flavo-treated cells are viable with negligible difference in growth rates (see revised Supp.Fig.10). Therefore, one would not expect a complete shut-

down of the transcriptional machinery in these cells. However, we do agree that at least some of the seemingly overexpressed genes in Fig.6B may just either have no change or less suppressed compared to the suppressed genes; we cannot tell without spike-in controls.

F,G: the authors should comment on why they did pre-treatment experiments. Reversal of kinase inhibitor effect (and hence re-expression of hypersensitive genes) should occur relatively quickly. How do the authors explain that the see effect duration up to several weeks?

Response: We did pre-treatment experiments as we could not subject the animals to flavopiridol treatments during immune checkpoint inhibitor therapy, because flavopiridol could, and probably will, affect the host immune functions as well.

Recovery from chronic flavopiridol seems to be cell-dependent. The p-Ser2 and p-Ser5 levels return to normal after 72 hours of release from flavopiridol in CT26 cells in vitro, although the recovery was significantly slower in B16F10 cells. See next:

However, the effect lasts significantly longer in vivo, and it can be seen even 3 weeks after animal injection (Supp.Fig.12b), which allowed us to successfully perform the in vivo studies here. These in vivo experiments were done 3 times with similar results each time. We believe that the relative instability of the phenotype in vitro may underlie the significant less incidence of the TEdeff phenotype among cancer cell lines compared to tumor tissues. We do not yet have enough data to explain this discrepancy between the in vitro and in vivo stability of this phenotype.

Reviewer #3 (Remarks to the Author):

This is a compressive study to describe the novel link between defective transcription elongation in cancers and treatment response. However, there are some questions the authors need to clarify and address:

1) For TCGA data, the authors did not provide sufficient information how isoform and gene level data were obtained (directly from TCGA level III?). About 10 cancer types were analyzed. What happened to other cancer types and how the authors decided to keep these in the paper? The number of samples used for each is also not clear.

Response: The gene- and isoform-level RNAseq data were obtained from GDAC firehose (so were the RPPA and clinical data); this is now explained in the Methods section. We reasoned that an analysis of ~10 cancers should be a sufficient demonstration of the wide-spread nature of this phenotype. However

now, we have included more datasets to a total of 27 cancer datasets from TCGA. The number of samples for each dataset/platform is now indicated in the Supplementary Table 1.

2) The authors evaluated processing methods, sample quality or batch for potential artifact but they were limited (to KIRC). What about mapping methods? Would tumor stroma affect it, particularly from immune cells?

Response: We have now included similar batch effect analyses for more cancers in Supp.Fig.2a. Batch-to-batch variations in gene-level and genome-level mRNA expression in RNAseq datasets in TCGA have been reported and are significant (<http://bioinformatics.mdanderson.org/tcgabatcheffects>). As such, it is not surprising that short-to-long isoform ratio of AT genes also shows some occasional variations in some cancers. Nevertheless, removing these “outlier” batches from the analyses presented in this study does not affect any of the conclusions (not shown).

Regarding the effect of mapping methods on the TEdeff phenotype: all of the processed RNAseq data from TCGA in the GDAC were processed using RSEM, which maps the reads directly onto the transcriptome. So there were no mapping method variations between the datasets or batches. Moreover, we obtained almost identical results using data processed using another method: kallisto (Supp.Fig.2b).

Regarding the effect of stroma/immune transcriptome: we considered this possibility early on. If this were the case, the TEdeff phenotype would have a very strong correlation with tumor purity and/or immune infiltration in tumor samples. However, although TEdeff correlated with higher immune cell infiltration in a few cases (KIRC, SKCM and BRCA), this was not consistent between cancer types, and the level of correlation was not strong enough to support this possibility.

3) How many cell lines from CCLE were analyzed? The defect event proportion appears much lower than the solid tumors (20%). How is this explained?

Response: We analyzed ~450 cell line data for which we had the RNAseq BAM files. To this list, we added an additional list of 40 lines based on our analysis of gene expression-level TEdeff signatures of the whole CCLE set (1,050 lines). The figure in Fig.3A reflects the short-to-long isoform ratios of this list of ~500 lines, out of which we identified and validated 4 (there were a few more that we did not test).

It is true that only a fraction of cell lines seems to show the TEdeff phenotype. We hypothesize that this is due to one or both of the following factors: although TEdeff is associated with the increased expression of cell cycle markers at mRNA and protein levels in tumor tissues (not shown), TEdeff lines (UACC-812 and MDA-MB-415) have a slow growth rate in vitro, possibly due to TEdeff-induced stress. It is well-known that the process of establishing cancer cell lines is a tedious one, and subjects the tumor cells to a substantial level of selection. Therefore, it is possible that TEdeff is selected against during such a process. Such a possibility is not without a precedent. For example, while malignant melanoma tissues show a wide range of aneuploidy levels, and a substantial portion with stable near-diploid genomes, all of the melanoma cell lines in CCLE are highly aneuploid (personal unpublished observations). Same goes for breast cancer and several other lineages as well, suggesting that the tumor cells with stable genomes are selected against during adaptation to the cell culture conditions.

Another possibility discussed above in response to another reviewer’s question is that the TEdeff phenotype may not be stable in vitro, as it is largely not genetically defined. For example, release from

chronic flavopiridol treatment causes the cells to revert their Pol II total and phospho-levels to initial state within a few days in CT26 cells. However, the Pol II defects seem to persist even after more than 3 weeks of in vivo growth in mice after flavopiridol release (Supp.Fig.12).

4) Fig G: how were the TEdeff vs. TEprof overall expression levels like? Did the authors normalize that for the plot?

Response: The reviewer did not indicate which figure, but it looks like this is about Fig.2G. The blue and red colors reflect lower or higher expression in TEdeff vs. TEprof samples of total or phospho-levels of the corresponding proteins. The RPPA data are normalized (by TCGA consortium) according to a rigorous normalization protocol taking into account overall intensities of all the assessed proteins for a given sample. We did not normalize the measures for overall protein levels further.

5) The authors mentioned mutations did not affect the finding but did they look at indel mutations (micro or short indels)? How about SNVs? Recent studies did report indel burden [PMID: 28694034] or type I interferon deletion [PMID: 29618619] affected immune treatment response.

Response: Yes, we tested indels, SNVs and CNVs. We did not see an enrichment of either of these among TEdeff tumors. In fact, as we reported in Fig.4, TEdeff could be combined with the mutational load for an even better separation of patients in terms of potential clinical benefits from immune checkpoint blockade.

6) I did not see the gene list with defective transcription elongation which should be provided in supplementary material.

Response: We now provide a list of the Class I and II genes in Supp.Table 2.

7) Reference for “kallisto” should be provided

Response: Fixed.

Reviewers' Comments:

Reviewer #1:

Remarks to the Author:

All of the reviewers comments have been addressed.

with respect to:

Figure 4A The graphs for TEdeff/prof and immunotherapy seem to be slightly different between the plots -why.

Response:

There are a few patients who were assigned both therapies (1 patient among TEdeff, and 7 patients among TEprof). We assigned them to one or the other in the bottom plots in Fig.4A, which creates small differences between the top and bottom plots. However, excluding these patients altogether does not change the pattern or the statistics significantly.

Please remove the patients that received double treatment.

Reviewer #2:

Remarks to the Author:

In this revised version of the manuscript, the authors have added new experimental evidence (such Pol II ChIP-seq and several rescue experiments) that support their initial conclusions.

In addition, the authors included additional description and interpretation that will help navigate the reader through the tremendous amount of data presented.

Several important points were clarified in the rebuttal letter. This relates especially to dose and timing of Flavo treatment.

Based on these changes, additions and clarifications, I support publication of the manuscript.

Reviewer #3:

Remarks to the Author:

The authors revised the manuscript adequately and I have no further comments.

Response to Reviewers

Reviewer #1 (Remarks to the Author):

Please remove the patients that received double treatment.

Response: We have removed patients in Fig.4A who were assigned both therapies. In addition, we redid the survival analyses for both KIRC and SKCM using the latest clinical survival data for TCGA compiled by Liu et al, Cell 2018.